

# Ground-based MAX-DOAS observations of tropospheric aerosols, NO₂, SO₂ and HCHO in Wuxi, China, from 2011 to 2014

Y. Wang[1], J. Lampel[1,2], P. H. Xie[3,4,5], S. Beirle[1], A. Li[3], D. X Wu[3], and T. Wagner[1]

[1] Max Planck Institute for Chemistry, Mainz, 55128, Germany

[2] Institute of Environmental Physics, University of Heidelberg, Heidelberg, 69120, Germany

[3] Anhui Institute of Optics and Fine Mechanics, Key laboratory of Environmental Optics and Technology, Chinese Academy of Sciences, Hefei, 230031, China

[4] CAS Center for Excellence in Urban Atmospheric Environment, Institute of Urban Environment, Chinese Academy of Sciences, Xiamen, 361021, China

[5] School of Environmental Science and Optoeclectronic Technology, University of Science and Technology of China, Hefei, 230026, China

*Correspondence to*: Y. Wang (y.wang@mpic.de); P.H. Xie (phxie@aiofm.ac.cn)

**Abstract.**

We characterize the temporal variation and spatial distribution of nitrogen dioxide ($NO_2$), sulphur dioxide ($SO_2$), formaldehyde (HCHO) and aerosol extinctions using vertical profiles derived from long-term Multi Axis - Differential Optical Absorption Spectroscopy (MAX-DOAS) observations from May 2011 to November 2014 in Wuxi, China. A new inversion algorithm (PriAM) is implemented to retrieve profiles of the trace gases (TGs) and aerosol extinction (AE) from the UV spectra of scattered sunlight recorded by the MAX-DOAS instrument. We investigated two important aspects of the

retrieval process. We found that the systematic seasonal variation of temperature and pressure (which is regularly observed in Wuxi) can lead to a systematic bias of the retrieved aerosol profiles (e.g. up 20% for the AOD) if it is not explicitly considered. In this study we take this effect for the first time into account. We also investigated in detail the reason for the differences of tropospheric VCDs derived from either the geometric approximation or by the integration of the retrieved profiles, which were reported by earlier studies. We found that these differences are almost entirely caused by the limitations

of the geometric approximation (especially for high aerosol loads). The results retrieved from the MAX-DOAS observations are compared with independent techniques not only under cloud free sky conditions, but also under various cloud scenarios. Under most cloudy conditions (except fog and optically thick clouds), the trace gas results still show good agreement. In contrast, from the aerosol results only near-surface AEs could be still well retrieved under cloudy situations.

After a quality controlling procedure, the MAX-DOAS data are used to characterize the seasonal, diurnal, and weekly

variations of $NO_2$, $SO_2$, HCHO and aerosols. A regular seasonality of the three trace gases is found, but not for aerosols. Similar diurnal variations are found for $SO_2$, HCHO and aerosols in different seasons, but not for $NO_2$. Similar annual variations of the profiles are found in different years, especially for the trace gases. Considerable amplitudes of weekly cycles occur for $NO_2$ and $SO_2$, but not for HCHO and aerosols. Good correlations between the TGs and aerosols are found,





especially for HCHO in winter. Significant wind direction dependencies of the trace gases, especially for the near-surface concentrations, are found, but only a weak dependence is found for aerosol properties, especially the AOD. Our findings imply that the local emissions from the industrial area (including traffic emissions) dominate the amount of local pollutants while long distance transport might also considerably contribute to the local aerosol levels.

**1 Introduction**

Nitrogen dioxide ($NO_2$), sulphur dioxide ($SO_2$), and formaldehyde (HCHO) are important atmospheric constituents which play crucial roles in tropospheric chemistry (Seinfeld and Pandis, 1998). $NO_2$ is involved in many chemical cycles such as the formation of tropospheric ozone. $NO_2$ and $SO_2$ can be converted to nitrate and sulfate through the reaction with the OH radical. HCHO is formed mainly from the oxidation of volatile organic compounds (VOCs) and methane. Primary emissions

of HCHO could be also important, especially in industrial regions (Chen et al., 2014). Due to the short life time of HCHO, it can be used as a measure of the level of the local VOC amount. The VOCs can then be eventually oxidized to form organic aerosols. $NO_2$, $SO_2$ and VOCs (marked by HCHO) are essential precursors of aerosols. During the industrialization and urbanization, anthropogenic emissions from traffic, heating, industry, and biomass burning have significantly increased the concentrations of these gases in the boundary layer in urban areas (Environmental Protection Agency, 1998; Seinfeld and

Pandis, 1998). Nowadays, strong haze pollution events occur frequently around megacities and urban agglomerations, especially in newly industrializing countries like China, and have a significant impact on human health (Fu et al., 2014a). Recent studies found that in megacities in different regions of China most of aerosol particles are formed through photochemistry of precursor gases during haze pollution events (Crippa et al., 2014 and Huang et al., 2014). Understanding the temporal variation and spatial distribution of the trace gases (TGs) and aerosols through long-term observations is thus

helpful to identify the dominating pollution sources, distinguish the contribution of transport and local emission as well as the relation between aerosols and their precursors. To accomplish this, one Multi Axis - Differential Optical Absorption Spectroscopy (MAX-DOAS) instrument was operated from 2011 to 2014 in Wuxi (China).

Since about 15 years, the MAX-DOAS technique has drawn lots of attention because of the potential to retrieve the vertical distribution of TGs and aerosols in the troposphere from the scattered sunlight recorded at multiple elevation angles

(Hönninger and Platt, 2002; Hönninger et al., 2004; Bobrowski et al., 2003; Wagner et al., 2004; Van Roozendael et al., 2003 and Wittrock et al., 2004) using relatively simple and cheap ground-based instrumentation. Ground based measurements of TG profiles are complementary to global satellite observations and allow for inter-comparisons and validation exercises (Irie et al., 2008; Roscoe et al., 2010; Ma et al., 2013; Kanaya et al., 2014; Vlemmix et al., 2015a). Using different inversion approaches, the column densities, vertical profiles and near-surface concentrations of the TGs and

aerosols can be derived and provide additional information compared to in-situ monitoring or satellite observations.

The tropospheric vertical column density (VCD) of TGs is either derived by the geometric approximation (e.g. Brinksma et al., 2008) or by integration of the retrieved concentration profiles (Vlemmix et al., 2015b). The near-surface concentration





can be derived using simplified rapid methods (Sinreich et al., 2013 and Wang et al., 2014b) or directly from the derived profile. The existing profile inversion schemes developed by different groups can be subdivided into two groups: the 'full profile inversion' based on optimal estimation (OE) theory (Rodgers, 2000; Frieß et al., 2006, 2011; Wittrock et al., 2006; Yilmaz, 2012; Clemer et al., 2010; Irie et al., 2008, 2011; Hartl and Wenig, 2013 and Wang et al., 2013a and b) and the so called parameterization approach using look-up tables (Li et al., 2010, 2012; Vlemmix et al., 2010, 2011; Wagner et al., 2011). In comparison with the look-up table methods, the OE-based inversion algorithms are in principle easily applied to different species, different measurement locations and instruments, but they require radiative transfer simulations during the inversion and can therefore be computationally expensive for large datasets. Clemer et al. (2010), Frieß et al. (2011), Kanaya et al. (2014), Hendrick et al. (2014), Wang et al. (2014a) applied their OE approaches to long-term MAX-DOAS observations in different locations of the world. The stability or flexibility of the inversion algorithms depends on the choice of the inversion approach, the iteration scheme and the a-priori constraints (Vlemmix et al., 2015b). Designing an approach balancing stability and flexibility is quite important for long-term observations because of the occurrences of various atmospheric scenarios caused by natural variability and human activities.

In this study, we use the Levenberg-Marquardt modified Gauss-Newton numerical procedure (Yilmaz, 2012) with some modifications to optimally balance stability and flexibility, which will be referred to in the following as "**Pr**ofile **i**nversion algorithm of aerosol extinction and trace gas concentration developed by **A**nhui Institute of optics and fine mechanics, Chinese academy of sciences (AIOFM, CAS) in cooperation with **M**ax Planck Institute for Chemistry (MPIC)" (PriAM) (Wang et al., 2013a and b). The PriAM algorithm joined the intercomparison exercise of aerosol vertical profiles retrieved from MAX-DOAS observations, between five inversion algorithms during the Cabauw Intercomparison Campaign of Nitrogen Dioxide measuring Instruments (CINDI) in summer 2009 (Frieß et al., 2016). The intercomparison displayed good agreements of the aerosol extinction (AE) profiles, AODs and near-surface AEs retrieved by the PriAM algorithm with those by other algorithms and with a collocated ceilometer instrument, a sun photometer and a humidity controlled nephelometer. In this work the PriAM is applied to the long-term MAX-DOAS observations in Wuxi, China. The retrieved results of $NO_2$, $SO_2$ and HCHO and aerosols are verified by comparisons with several independent data sets for a period longer than one year.

Under cloudy skies the retrieval algorithm could be subject to large errors because of the increased complexity of the atmospheric light paths inside clouds (e.g. Erle et al., 1995; Wagner et al., 1998, 2002, 2004; Winterrath et al., 1999), which are usually not considered in the forward model. Previous studies usually simply discard cloud-contaminated measurements. However, depending on location and season, a large fraction of measurements might be affected by clouds, e.g. about 80% of all MAX-DOAS measurements in Wuxi (Wang et al., 2015). We investigate the effect of clouds on the different MAX-DOAS retrieval results of aerosols and TGs, especially the near-surface concentrations by comparisons with results from independent techniques under various cloud scenarios. Information on different cloud scenarios is directly derived from the MAX-DOAS observations and can thus be assigned to each MAX-DOAS result without temporal interpolation. Tropospheric TG VCDs are also important for satellite validation. So far, most studies used the so called geometric



approximation to derive TG VCDs from MAX-DOAS measurements. However, considerable systematic discrepancies of tropospheric TG VCDs derived by the geometric approximation and by integration of the TG profiles are already reported in Hendrick et al. (2015), but which of the two values is closer to reality remains unclear. It is essential to answer this question in order to use a trustworthy method to determine the tropospheric TG VCDs. In this study we show evidence that the

5 dominant error is associated with the geometric approximation, thus the TG VCDs by integration of the profiles are used for further studies here. After the series of verification exercises, the MAX-DOAS results are used to characterize temporal variations and vertical distributions of aerosols and TGs in Wuxi. The relation between aerosols and TGs are also discussed.

The paper is organized as follows: in section 2 the observations and different steps of the data analysis are described and

10 results are verified. Moreover, the cloud effect on the retrievals and the errors of the geometric approximation are discussed. In section 3 we characterize seasonal variations and inter-annual trends, diurnal variations, weekly cycles and wind dependencies of the aerosols and TGs. The relation between aerosols and TGs are also discussed. In section 4 the results are discussed and conclusions are given.

## 2 MAX-DOAS measurements

### 2.1 MAX-DOAS in Wuxi station

A MAX-DOAS instrument developed by AIOFM shown in Fig. 1a is located on the roof of a 11-story building in Wuxi City (Fig. 1b), China (31.57 °N, 120.31 °E, 50 m a.s.l.) at the transition between the urban and suburban area. The suburban area with lots of farmlands is located in the east, and Taihu Lake is located in the north. The heavily industrialised area and the urban centre (living and business area) are in southwest and northwest direction of the MAX-DOAS station, respectively.

Wuxi city belongs to the Yangtze River delta industrial zone and is located about 130 km north-west of Shanghai (Fig. 1c). Wuxi is an important industrial city and has about six million inhabitants. Because of the high population density and high industrial activity, relatively high abundances of $NO_2$, $SO_2$ and VOCs are found (Fu, et al, 2013). Fig. 1d displays the mean distributions of $NO_2$ (Boersma et al., 2011), HCHO (de Smedt et al., 2010) and $SO_2$ (Theys et al., 2015) as derived from the Ozone Monitoring instrument (OMI) (Levelt et al., 2006 b). In north-west direction of Wuxi city the large industrial zone of

North China plain is located, which has even higher pollution loads.

The MAX-DOAS instrument was operated by the Wuxi CAS Photonics Co. Ltd from May 2011 to December 2014. The instrument was pointed to the north and automatically recorded spectra of UV scattered sunlight at sequences consisting of five elevation angles (5 °, 10 °, 20 °, 30 ° and 90 °). One elevation sequence scan took about 12 min depending on the received radiance. More details of the instrument can be found in Wang et al. (2015). During the whole observation period, the

30 instrument stopped twice: 15 December 2012 to 29 February 2013 and 16 July to 12 August 2013.



## 2.2 Retrievals of the tropospheric profiles of aerosol extinctions, $NO_2$, $SO_2$ and HCHO volume mixing ratios.

### 2.2.1 Retrieval of slant column densities

The slant column densities (SCDs) of the oxygen dimer ($O_4$), $NO_2$, $SO_2$ and HCHO are retrieved from scattered sunlight spectra measured by the MAX-DOAS instrument using the DOAS technique (Platt and Stutz, 2008) implemented by the WINDOAS software (Fayt and van Roozendael, 2009). SCD represents the TG concentrations integrated along the effective atmospheric light path. The TG cross sections, wavelength ranges and additional properties of the DOAS analysis are provided in Table 1. Fig. 2 shows typical DOAS fit examples. We skip data for SZAs larger than 75 ° because of stronger absorptions of stratospheric species and low signal to noise ratio. We also skip the data with large root mean square (RMS) of the residuals and large relative intensity offset (RIO). All thresholds of the quantities used for filtering the results and the percentages of screened data of the total number of observations are listed in Table 2. Detailed discussions of the DOAS fit parameters for each species can be found in section 1 of the supplement.

### 2.2.2 The PriAM algorithm

In the first step of the retrieval, tropospheric vertical profiles (in the layer from the ground to the altitude of 4 km) of aerosol extinction are retrieved from the $O_4$ dSCDs. Afterwards, the profiles of $NO_2$, $SO_2$ and HCHO volume mixing ratios (VMRs) are retrieved from the respective dSCDs in each MAX-DOAS elevation angle sequence by using the PriAM algorithm, which is described in Wang et al., 2013a and b, both in Chinese language. We summarize the basic concept of the PriAM algorithm and its implementation settings for this study below, while details can be found in the section 2 of the supplement. In PriAM the retrieval problem is solved by the Levenberg-Marquardt modified Gauss-Newton numerical iteration procedure (Rodgers, 2000). Considering the frequent variation of aerosols and the TGs, very little is known about the expected profiles. Thus a set of fixed a-priori profiles is used for each species. A smoothed box-shaped a-priori AE profile (Boltzmann distribution) (Yilmaz, 2012), exponential a-priori profiles of $NO_2$ and $SO_2$ (similar to Yilmaz, 2012 and Hendrick et al., 2014), and a Boltzmann distribution a-priori HCHO profile (based on the MAX-DOAS and aircraft measurements in Milano during summer of 2003 reported in Wagner et al., 2011) are used by the PriAM algorithm and denoted by the grey curves in Fig. 7, respectively. Besides these standard a-priori profiles, we tested the effect of changing the profile shape and absolute value on the fit results. The description of these sensitivity tests is provided in section 2.1 of the supplement. We conclude that the standard a-priori profiles are an optimum choice for the application to the long term MAX-DOAS measurements in Wuxi. We also find that improper a-priori profiles can strongly impact the aerosol profile retrievals, but only slightly impact the TG results.

PriAM uses the radiative transfer model (RTM) SCIATRAN 2.2 (Rozanov et al., 2005). Based on the wavelength intervals of the DOAS fit, the RTM simulations are done at 370 nm for the retrieval of aerosols and $NO_2$, at 339 nm for HCHO and at 313 nm for $SO_2$. The surface height and surface albedo are set as 50 m a.s.l. and 0.05, respectively. The single scattering albedo (0.9±0.05) and asymmetry factor (Henyey and Greenstein, 1941) (0.72±0.03) are chosen according to inversion





results from the Taihu AERONET station from 2011 to 2013 (the data in 2014 is unavailable). The retrieved aerosol extinction at 370 nm is converted to those around 313 nm for the $SO_2$ and 339 nm for the HCHO retrieval using Ångström exponents derived also from the Taihu AERONET data sets.

In addition, here it should be noted that, the Levenberg-Marquardt modified Gauss-Newton procedure is based on the
assumption that the probability distribution function (pdf) of the atmospheric state (x) can be described by a Gaussian pdf (P) around the a-priori state (xa) (Rodgers, 2000):

$$-2 \ln P(x) = (x - x_a)^T S_a^{-1} (x - x_a) + c \tag{1}$$

Here c is a constant value and $S_a$ is the covariance matrix of the a-priori. Thus the solution can not reach the true state when the pdf of the atmospheric state (x) is skew or asymmetric (Rodgers, 2000). In this study the retrieval of the AE for
extremely high aerosol loads (e.g. fog and haze) belongs to cases, which do not fulfil this assumption. In such cases the AE is underestimated by the inversion (see session 2.2.6).

### 2.2.3 Correcting the effect of the variation of ambient temperature and pressure

In previous studies (Clemer et al., 2010; Hendrick et al., 2014; Wang et al. 2014a) usually fixed temperature and pressure (TP) profiles are used (e.g. obtained from the US standard summer atmosphere for the measurements in China). However for
locations with a significant and systematic annual variation of TP, as in this study, this simplification can affect the retrieved AODs and AE profiles (and thus also the TG profiles) systematically, yielding virtual seasonal variations. The time series of TP near the surface from the weather station nearby the MAX-DOAS instrument are shown in Fig. 3 for the year 2012 (similar patterns are found for other years, see Fig. S10 of the supplement). A regular annual variation of surface TP is obvious with amplitudes between winter and summer of about 20 K and 30 hPa, respectively. The $O_4$ VCDs derived from
the fitted curves of surface TP (the method is described in the section 3 of the supplement) is also shown in Fig. 3. The $O_4$ VCD in summer is systematically lower than in winter by about 15% of the yearly mean $O_4$ VCD. Ignoring this systematic seasonal variation can cause a 20-30% bias of the AOD and near-surface aerosol extinction (see details in section 3 of the supplement). The error of the aerosol retrieval can further nonlinearly impact the TG profile retrievals. To account for this effect, the seasonal variation of TP and the $O_4$ VCD is parameterized and explicitly considered in the forward model during
the MAX-DOAS retrievals by the PriAM algorithm. Figure 4 shows the AOD retrieved by PriAM using either explicit TP information or the TP profiles from the US summer standard atmosphere. The consistency of the AOD retrieved based on the explicit TP data with the simultaneous Taihu AERONET level 1.5 AOD data sets (see section 2.2.5) is better than for TP profiles from the US standard summer atmosphere.

### 2.2.4 Evaluation of the internal consistency of the inversion algorithm

In this section the retrieval quality is evaluated for favourable measurement conditions, namely cloud-free sky with relatively low aerosols (average AOD of about 0.6), and the performance of the retrievals in different seasons is discussed.



Comparing the measured TG dSCDs to the modelled dSCDs (the results of the forward model corresponding to the retrieved AE and TG profiles) is a direct way to evaluate how close to the real profile the retrieved profile is. Ideally, the differences between measured and modelled dSCDs are minimized by the inversion. However because of measurement errors, deviations of the forward model from reality (e.g. for cloudy skies, shown in section 2.2.6) and the not always realistic

assumption of the Gauss-Newton Algorithm in Eq. (1) (especially under the condition with strong aerosol load, also shown in section 2.2.6), the derived profiles might strongly deviate from the real profiles. Figure 5 shows the mean differences (and standard deviations denoted by error bars) between the measured and modelled dSCDs for the four species during the whole measurement period. Almost symmetrical Gaussian-shape histograms of the absolute differences for the different elevation angles are found and shown in Fig. S11 of the supplement. For the aerosol retrieval, a larger negative difference of the $O_4$

dSCD of $2.9 \times 10^{41}$ molecules$^2$ cm$^{-5}$ is found for 5 °elevation angle, indicating an underestimation of the aerosol extinction in the layer close to the surface; however the magnitude of the underestimation is only about 2% based on the mean $O_4$ dSCD of about $1.6 \times 10^{43}$ molecules$^2$ cm$^{-5}$ for 5 °elevation angle. For the TG retrievals, in general the differences for high elevation angles are slightly larger than those for low elevation angles. This finding probably indicates the higher sensitivity of the inversion algorithm to lower altitudes. This is also indicated by the averaging kernels in Fig. 6b. Even so, the mean

deviations of the dSCDs for the 30 ° elevation angle are only $-0.28 \times 10^{15}$ molecules cm$^{-2}$ for NO$_2$ (mean dSCD of $2.6 \times 10^{16}$ molecules cm$^{-2}$), $-0.07 \times 10^{15}$ molecules cm$^{-2}$ for SO$_2$ (mean dSCD of $3.3 \times 10^{16}$ molecules cm$^{-2}$) and $0.65 \times 10^{15}$ molecules cm$^{-2}$ for HCHO (mean dSCD of $1.6 \times 10^{16}$ molecules cm$^{-2}$).

The mean averaging kernels (AKs) for retrievals of AE and the NO$_2$ VMR are shown in Fig. 6. AKs for SO$_2$ and HCHO are similar to NO$_2$ (see Fig. S14c and S15c of the supplement). They indicate that the inversions are sensitive to the layers from

20 the surface up to 1.5 km. The degrees of freedom (DoF) are about 1.5 for aerosols (similar to Frieß et al., 2006), 2 for NO$_2$ and 2.3 for SO$_2$ and HCHO. AKs are generally similar for different seasons (see Fig. S12d -S15d of the supplement), indicating the consistent response of the measurements to the true atmospheric state. The slight seasonality is probably related to the variation of the SZA. The same reason probably causes the weak diurnal variation of the DoF of the inversions shown in Fig. S16 of the supplement. The averaged profiles retrieved from the measurements during the whole period and in

different seasons are shown in Fig. 7 together with the corresponding a-priori profiles. The retrieved profiles below 1.5 km are quite different from the a-priori profiles, indicating that the measurements contain sufficient information for the altitude below 1.5 km. The mean contributions of the noise and the smoothing error (this error originates from the limited resolution of the inversion) of the retrievals are shown in Fig. S12b - S15b of the supplement. The total (absolute) retrieval errors have a maximum around 1 km and decrease towards the surface. The relative errors are minimal close to the surface (10% for AE,

NO$_2$ and SO$_2$, and 30% for HCHO). Most of the errors originate from the smoothing error, which largely contributes to the total error at high altitudes.




### 2.2.5 Comparisons with independent data sets under clear skies

To validate the results from MAX-DOAS observations, the column densities and averaged concentrations in the lowest layer from 0 to 200m are compared to independent measurements:

(a)  AODs at 380 nm (level 1.5) from the sun photometer at the AERONET (Holben et al., 1998 and 2001) Taihu station. The data is downloaded from the website of http://aeronet.gsfc.nasa.gov/. The AERONET sun photometer is located 18 km south west of the MAX-DOAS instrument. AERONET data in the period from May 2011 to October 2013 is included in the study. In the level 1.5 data, a cloud screening scheme is used to filter most of the cloud contaminated data (Smirnov et al., 2000).

(b)  Visibilities near the ground from a forward-scattering visibility meter (Manufacturer: Anhui Landun Photoelectron Co. Ltd. Model: DNQ2 forward-scattering visibility meter) (Wang et al., 2015), which is located at the same site as the MAX-DOAS instrument. The data from May 2011 to December 2013 is available.

(c)  $NO_2$ and $SO_2$ VMRs (no HCHO data are available) near the ground from a long path DOAS (LP-DOAS) instrument (Qin et al., 2006) located at the same site as the MAX-DOAS instrument. The LP-DOAS is directed to the East with a total light path length of about 2km. The data from May 2011 to April 2012 is available

MAX-DOAS results are compared to the available independent measurements within 15 minute time difference.

In this section only the data recorded during clear sky conditions with low aerosol load are compared to the MAX-DOAS results (comparisons for different cloud conditions are shown in the section 2.2.6). Almost symmetrical Gaussian-shape histograms of the absolute difference of the AODs from MAX-DOAS and AERONET for different seasons except summer are found and shown in Fig. S17a of the supplement. The averaged absolute differences and standard deviations (indicated by the error bars) of the AODs are shown in Fig. 8a. The mean differences are smaller than 0.16. The AODs from MAX-DOAS and AERONET show correlation coefficients (Pearson's product moment correlation coefficient is applied in this paper) within 0.56 to 0.91 (see Fig 9). The highest coefficient of 0.91 is found in summer, probably related to the wider range of AODs covered, but in that season also the largest absolute difference of -0.16 is found probably due to the stronger aerosol load than in other seasons. Underestimation of high aerosol amounts by MAX-DOAS will be discussed in session 2.2.6. In spring, there are several points (mostly in May of 2011 and 2012) above the 1:1 line. For this finding we have currently no explanation. Several previous studies applied a correction factor to measured $O_4$ dSCDs to improve the consistency between the AODs derived from MAX-DOAS and those from AERONET (e.g. Wagner et al., 2009; Clemer et al., 2010 and Frieß et al., 2016). And so far there is no credible explanation for this correction factor. In this study we don't apply any correction factor, because we achieve reasonable consistency between MAX-DOAS and AERONET results without the application of a correction factor.

The averaged AEs in the lowest layer derived from the MAX-DOAS are compared with those from the visibility meter. Here it has to be noted that both instruments do not probe exactly the same air masses: the visibility meter is sensitive to air masses at the measurement location while the MAX-DOAS is sensitive to the air masses along the line of sight for up to





several kilometres away from the instrument and up to a few hundred meters above the ground. Fig. S17b of the supplement shows almost symmetrical Gaussian-shape histograms of the absolute differences of the AEs between the two techniques. The mean differences are $< 0.18$ km$^{-1}$ as shown in Fig. 8b. The highest correlation coefficient of 0.74 is found in summer probably related to the wider range of values and the stronger vertical convection, which causes a higher boundary layer and

5 possibly a smoother vertical distribution of aerosols than in other seasons (see Fig. 10). In spring, the worst correlation is found and might be related to the occurrence of long-distance transport of dust with elevated aerosol layers (see section 3.2). The VMRs of $NO_2$ and $SO_2$ in the lowest layer derived from MAX-DOAS are compared with the values from LP-DOAS measurements for the individual seasons. Like for the AE, it has to be noted that both instruments do not probe exactly the same air masses; as the LP-DOAS yields the mean TG concentration for the light path defined by the set-up of instrument

and reflector. In general the mean absolute differences are smaller than 5 ppb for $NO_2$ and 6 ppb for $SO_2$ (see Fig. 8c). Almost symmetrical Gaussian-shape histograms of the absolute differences are also found for $NO_2$ and $SO_2$ in different seasons (Fig. S17c and d of the supplement). The correlation coefficients range from 0.4 to 0.7 for $NO_2$ (see Fig. 11) and from 0.7 to 0.8 for $SO_2$ (see Fig. 12) in all seasons. The higher correlation coefficients for $SO_2$ than for $NO_2$ are probably related to the longer lifetime and thus more homogeneous vertical and horizontal distribution of $SO_2$ compared to $NO_2$,

especially in the layer from 0 to 200m. The worst correlation of $NO_2$, especially in the afternoon (see Fig. S18 of the supplement) is found in summer probably because of the low $NO_2$ VMR near the surface, the small value range and the steep vertical gradient in the layer from 0 to 200m (see below). The generally positive absolute differences of $NO_2$ and $SO_2$ shown in Fig. 8c and d could be attributed to strong gradients in the layer from 0 to 200m as e.g. found from tower measurements in Beijing, Meng et al. (2008): they concluded that the largest values of the $NO_2$ and $SO_2$ concentrations are not directly located

at the surface, but at an altitude of about 100 meters, especially in summer. However, it should be noted that the vertical gradients around Wuxi might be different from those in Beijing and thus also other reasons might contribute to the observed differences.

### 2.2.6 Evaluations of retrievals under cloudy and strong aerosol conditions.

The retrieval of AEs by PriAM from $O_4$ absorptions is based on a forward model, which does not include the effects of

25 clouds. In principle it should be possible to also include cloud effects in the forward model (at least for horizontally homogenous clouds), but in the current version of our retrieval this is not yet accomplished. In this section, we investigate how strongly different types of clouds affect the MAX-DOAS retrieval results of aerosols and TGs. For that purpose we compare the MAX-DOAS results with independent data sets for different cloud types. For the characterization of the cloud conditions we use the cloud classification scheme described in Wang et al., 2015 (based on the concept of Wagner et al.,

2014) to classify the sky conditions from the MAX-DOAS observations, i.e. radiance, colour index and $O_4$ absorption. The scheme differentiates between eight primary sky conditions (varying between clear skies with low aerosol load to continuous cloud cover) and two secondary sky conditions of fog and optically thick clouds. In this study we condense the eight primary sky conditions to five primary conditions by merging two types of cloud holes and two types of continuous clouds and





ignoring the rare condition of "extremely high midday CI" (Wang et al., 2015). The remaining five primary conditions are clear sky with low aerosol loads ("low aerosols"), clear sky with high aerosol loads ("high aerosol"), "cloud holes", "broken clouds", and "continuous clouds". Each MAX-DOAS measurement scan is assigned to one of the five primary sky conditions. In addition, they can be assigned to the two secondary sky conditions of "fog" and "optically thick clouds". Here

it should be clarified that the "fog" sky condition does not exactly belong to the meteorology definition, but represents a sky condition derived from MAX-DOAS observations with a low visibility. In addition to the cloud effect, also the effect of high aerosol loads is evaluated (due to the unrealistic assumption of the pdf of the atmospheric state in the OE algorithm for high aerosol loads (see Eq. (1)).

Firstly measured and modelled dSCDs (results of the forward model) are compared under various sky conditions. In Fig. 13

(grey columns), the histograms of the differences between the measured and modelled dSCDs are shown for the four species (note Fig. 13 represents the differences for all non-zenith elevation angles). The histograms are symmetric and the maximum probabilities occur around zero for all four species. I.e., overall, there is no indication for a significant systematic retrieval bias. In the same figure, the relative frequencies for the different sky conditions are shown in different colours. In general, for cloudy sky conditions, especially for continuous clouds and optically thick clouds, larger discrepancies are found

compared to cloud free sky conditions. The effect of clouds on the inversion is stronger for aerosols than for TGs. For the aerosol inversion, more negative differences are found for "fog", which indicates that the strong extinction in "fog" is not well represented by the forward model (The phenomenon is also found in Fig. 15 and discussed below). To skip those inverted profiles, which probably differ largely from the real profiles, we only keep the profiles, for which the differences between measured and modelled dSCDs are smaller than $2 \times 10^{42}$ molecules$^2$ cm$^{-5}$ for the O$_4$ dSCDs (90.6% of the total

observations) and $5 \times 10^{15}$ molecules cm$^{-2}$ for NO$_2$ (89.8%), SO$_2$ (90.4%), and HCHO dSCDs (97.9%) for each elevation angle in one elevation sequence.

After this screening of potentially bad profiles, the mean profiles of AEs and TG mixing ratios as well as the corresponding total averaging kernels (which represent the sum of the averaging kernels at the individual altitudes) are shown in Fig. 14 for different sky conditions. While the total averaging kernels differ only slightly, the resulting profiles are quite different for

different sky conditions. There are two interesting findings for the retrieved profiles: first, for all cloudy scenarios (incl. fog), the maximum AE is not found at the surface, but at higher altitudes, as observed also by Nasse et al., (2015). This can be explained by the fact that clouds act as a diffusing screen. The effect on MAX-DOAS observations is that the light paths, especially for low elevation angles, become longer than for cloud-free conditions. Consequently, also increased O$_4$ absorptions are measured for such conditions. A similar effect can also be caused by elevated aerosol layers. Since the

forward model does not explicitly include clouds, usually elevated 'cloud-induced' aerosol layers are derived in the profile inversion under cloudy conditions. The diffusing screen effect depends on the cloud optical thickness. The most pronounced cloud-induced elevated aerosol layers are retrieved for optically thick clouds.

Interestingly, also for measurements under "fog" conditions, elevated aerosol layers are obtained from the MAX-DOAS inversion. This is at first sight surprising, but can be explained by two aspects: first, for most measurements classified as





"fog", still a systematic dependence of the $O_4$ dSCDs on elevation angles is found, indicating that during most "fog" events the visibility is still not close to zero. Second, for most of the measurements classified as "fog" also the presence of clouds (including thick and broken clouds) was detected (Wang et al., 2015). This finding indicates that, for most observations classified as "fog", increased aerosol scattering close to the surface occurred indeed, but at higher altitudes, even larger extinction was present. We also found a general larger value of the cost function under cloudy conditions (consistent with Fig. 13) and a systematic variation of the TG VCDs and near-surface VMRs for the different cloud scenarios. Besides measurement errors, these variations are probably also due to different photolysis rates and atmospheric dynamics (see Fig. S19 in the supplement).

In the following we compare the results from MAX-DOAS and other techniques under different sky conditions. Since the frequencies of different cloud conditions depend on season (Wang et al., 2015) and also the agreements between MAX-DOAS and other techniques were found to be different for different seasons (see section 2.2.5), the comparisons are done for individual seasons. In Figs. 15 to 19 the comparison results for autumn are shown (similar conclusion are found for other seasons and the relevant figures are shown in Fig. S20 – S23 of the supplement). Based on the comparisons of the retrieved profiles for different sky conditions (Fig. 14) and the comparison results with independent data sets (Figs. 15 – 19), we have developed recommendations, under which sky conditions which data product might be still useful or should better not be used. These recommendations, as summarized in Table 3, should not be seen as generally binding, but rather as a general indication of the usefulness of a given observation, and might change for improved inversion algorithms in the future.

In general we find that the aerosol results are more strongly affected by the presence of clouds. This is especially true for the retrieved AOD. Thus we recommend that retrieved AOD and AE profiles (except close to the surface) should not be used for all cloudy conditions. However, AE close to the surface can still well be retrieved under most cloudy conditions (except for thick clouds or fog). The TG results are less affected by clouds. Thus not only surface mixing ratios, but also TG profiles and tropospheric VCDs can still be well retrieved for most cloudy situations (except for thick clouds and fog). The MAX-DOAS data used in Section 3 are filtered by the recommendations listed in Table 3.

### 2.2.7 Error budgets

For the MAX-DOAS results, we derive the error estimates from different sources. Firstly we estimate the error budgets for the near surface values and column densities of the TGs and aerosols, which are summarized in Table 4. The following error sources are considered:

(a) Smoothing and noise errors (fitting error of DOAS fits) on the near-surface values and column densities are derived from the averaged error of profiles from the retrievals (shown in Fig. S12b - S15b of the supplement), and amount on average to 10% and 6% for aerosols, 12% and 17% for $NO_2$, 19% and 25% for $SO_2$ and 50% and 50% for HCHO, respectively.

(b) Algorithm errors related to an imperfect minimum of the cost function, namely the discrepancy between the measured and modelled dSCDs. Based on the fact that measurements for 5° and 30° elevation angles are sensitive to the low and





high air layers, respectively, we estimate the algorithm errors on the near-surface values and the column densities using the averaged relative differences between measured and modelled dSCDs for 5 ° and 30 ° elevation angle, respectively. These errors on the near-surface values and the column densities are on average estimated at 4% and 8% for aerosols, 3% and 11% for $NO_2$, 4% and 10% for $SO_2$, 4% and 11% for HCHO, respectively.

(c)   Cross section errors of $O_4$ (aerosols), $NO_2$, $SO_2$, and HCHO are 5%, 3%, 5% and 9%, respectively according to Thalman and Volkamer (2013), Vandaele et al. (1998), Bogumil et al. (2003) and Meller and Moortgat (2000).

(d)   The errors related to the temperature dependence of the cross sections are estimated in the following way. We firstly calculate the amplitude changes of the cross sections per kelvin using two cross sections at two temperatures from the same data sets. Then the amplitude changes per kelvin are multiplied by the variation magnitude of the ambient temperature (45 k during the whole measurement period, see Fig. 3). The corresponding systematic error of $O_4$ (aerosols), $NO_2$, $SO_2$ and HCHO are estimated to up to 10%, 2%, 3% and 6%, respectively.

(e)   The errors of TGs related to the errors of aerosols are estimated at 16% for VCDs and 15% for near-surface VMRs for the three TGs according to the total error budgets of aerosol retrievals.

The total error budgets on the TGs and aerosols are given by combining all the above error sources in the bottom row of Table 4. In general the sum of the smoothing and noise error is the dominant error source in the total error budget.

The error budgets of the profiles also consist of the five (four for aerosol profiles) error sources. The error (a) depends on the height, has much larger (relative) error at high altitudes and is already shown in Fig. S12b - S15b of the supplement. The error (b) can not be realistically estimated because of the difficulty of assigning discrepancies between measured and modelled dSCDs to each altitude of profiles. The error (c) and (d) have the identical number at all the altitudes and are same as the estimations for the near surface values and column densities above. The error (e) of TG profiles can be estimated as the total error budgets of aerosol profiles. However because of error (b) is unknown, the error (e) can not be quantified at the moment.

## 2.3 Comparisons between geometrical VCD and VCD from profile inversion

The geometric approximation (e.g. Brinksma et al., 2008) is often used to convert the dSCD for an elevation angle of α ($dSCD_\alpha$) to the tropospheric $VCD_{geo}$:

$$VCD_{geo} = \frac{dSCD_\alpha}{\frac{1}{sin(\alpha)} - 1} \tag{2}$$

The elevation angles between about 30 ° and 20 ° are usually used for the application of the geometric approximation (e.g., Ma et al., 2013 and Shaiganfar et al., 2011). The tropospheric VCD ($VCD_{pro}$) can also be derived by the vertical integration of the retrieved profiles. The relative differences ($Diff_{total}$) between $VCD_{pro}$ and $VCD_{geo}$ for $NO_2$, $SO_2$ and HCHO are calculated by Eq. (3):

$$Diff_{total} = \frac{VCD_{geo} - VCD_{pro}}{VCD_{pro}} \tag{3}$$



In Fig. 20, the average relative differences for elevation angles of 30 ° and 20 ° are shown as function of the relative azimuth angle (RAA), i.e. the difference between the azimuth angles of the sun and the viewing direction of the telescope. In general, the discrepancy is larger for an elevation angle of 30 ° than for 20 °. In addition, also an increase of the difference with increasing relative azimuth angle is found. Both findings have different magnitudes for the different TGs. The observed dependencies could be attributed to two reasons: first, the validity of the geometric approximation is limited, especially if the last scattering event occurs in the TG layer of interest. The respective probability depends on the layer height, wavelength, aerosol load and viewing geometry. A second reason for the observed differences is the uncertainty of the profile inversion. Some studies already reported systematic errors of the geometrical approximation:

1) Ma et al. (2013) showed that the systematic error of the $NO_2$ VCDs calculated by the geometrical approximation for an elevation angle of 30 ° is about 20% on average, which is quite similar with the value in Fig. 20b. Also, the error is larger for larger elevation angles and larger RAA, which is also consistent with the results shown in Fig. 20a and b.

2) The simulation studies for an elevation angle of 22 ° in Shaiganfar et al. (2011) show that the error of the geometrical approximation depends on the layer height of the TGs and aerosols. They found that a higher layer of TGs leads to a larger negative error. This finding is consistent with the results shown in Fig 20e, where the largest biases are found for HCHO, which has a higher layer than $NO_2$ and $SO_2$ (see Fig. 7)

To identify the dominating error source, we split the total difference ($Diff_{total}$) between $VCD_{geo}$ and $VCD_{pro}$ into two parts: The first part is the difference between $VCD_{geo}$ and $VCD_{geo}^m$. Here $VCD_{geo}^m$ is calculated by applying the geometric approximation to the modelled dSCD (from the forward model of the profile inversion) for the same elevation angle. This difference describes the error from the profile inversion and is referred to as $Diff_{inversion}$:

$$Diff_{inversion} = \frac{VCD_{geo} - VCD_{geo}^m}{VCD_{pro}} \qquad (4)$$

The second part is the difference ($Diff_{geometry}$) between $VCD_{geo}^m$ and $VCD_{pro}$:

$$Diff_{geometry} = \frac{VCD_{geo}^m - VCD_{pro}}{VCD_{pro}} \qquad (5)$$

$Diff_{geometry}$ describes the error due to the limitations of the geometric approximation. $Diff_{inversion}$ and $Diff_{geometry}$ are also shown in Fig. 20 with red and blue colours, respectively. It is found that $Diff_{inversion}$ is mostly smaller than 4% for the 30 ° elevation angle of and smaller than 2% for the 20 ° elevation angle. Moreover, the variation of $Diff_{total}$ along RAA is similar with $Diff_{geometry}$. Both findings clearly indicate that the error due to the limitation of the geometric approximation is the dominating error contributing to $Diff_{total}$. Moreover the systematic errors of the geometric approximation become significant when the aerosol load is large (see the section 4 of the supplement). Thus in the following, we integrate the retrieved profiles to extract the respective tropospheric VCD.



## 3 Results and discussion

In this section, MAX-DOAS results of column densities, near-surface concentrations and vertical profiles of aerosols and TGs are shown and discussed for a) seasonal variations and inter-annual trends, b) diurnal variations, c) weekly cycles as well as wind dependencies.

### 3.1 Meteorological conditions

The ground based weather station near the MAX-DOAS instrument records the ambient temperature, wind speed and direction, and relative humidity during the whole observation period. Fig 21 shows their seasonally mean diurnal variations. A large seasonal difference occurs only for the ambient temperature, but not for the wind speed and relative humidity. Similar diurnal variations for the three meteorology parameters are found for the different seasons. The ambient temperature and the relative humidity reach the maximum and minimum values around noon, respectively. The wind speed has the maximum value around 16:00 LT. The wind directions recorded by the same weather station are shown by the wind roses for the individual seasons in Fig. 22, indicating that the dominant wind is from the northeast in all seasons. In spring and summer the non-dominant wind directions occur more frequently than in winter and autumn.

### 3.2 Seasonal variations and inter-annual trends of daytime $NO_2$, $SO_2$, HCHO and aerosols

The time series of monthly averaged (after daily averaging) TG VCDs and near-surface VMRs as well as AODs and near-surface AEs (all the data are filtered by the recommended scheme in Table 3) derived from MAX-DOAS observations are presented in Fig. 23. Also shown are AODs and AEs obtained from AERONET and visibility meter, respectively.

Similar annual variations are found for TG VCDs and near-surface VMRs. The seasonal cycles of $NO_2$ and $SO_2$ show minimum values ($NO_2$ and $SO_2$ VCD of 9-17$\times10^{15}$ and 12-23 $\times10^{15}$ molecules cm$^{-2}$, respectively; $NO_2$ and $SO_2$ VMR of 5-11 and 4-11 ppb, respectively) in summer and maximum values ($NO_2$ and $SO_2$ VCD of 27-35$\times10^{15}$ and 33-54 $\times10^{15}$ molecules cm$^{-2}$, respectively; $NO_2$ and $SO_2$ VMR of 12-16 ppb and 14-18 ppb) in winter. These characteristics are already well-known over urban areas in the eastern China region (Richter et al., 2005, Ma et al., 2013; Hendrick et al., 2014, Qi et al, 2012 and Wang et al., 2014a). In contrast, HCHO shows an opposite seasonality compared to $NO_2$ and $SO_2$. The HCHO VCD and near-surface VMR are 16-20$\times10^{15}$ molecules cm$^{-2}$ and 4-6 ppb in summer, respectively, 7-10$\times10^{15}$ molecules cm$^{-2}$ and 2-4 ppb in winter, respectively. A similar seasonality of HCHO in the eastern China region was already reported by De Smedt et al., (2010 and 2015).

For AOD and AE no pronounced seasonal cycle is found. The MAX-DOAS results mostly reveal similar levels like the other two techniques. Note that the data in 2014 is not available from both the AERONET Taihu station and the visibility meter. The AOD is typically larger than 0.7 and the AE typically larger than 0.5 km$^{-1}$. Note that the extremely low values in July and August of 2013 are unrepresentative because of low statistics caused by the temporal shutdown of the instrument (see Fig. 23c).



The observed seasonal variations of the different species are related to three factors: the seasonal variation of emissions (or chemistry formation mechanism), removal mechanisms, and atmospheric transports (Wang et al., 2010; Lin et al., 2011). Different from the column densities, the near-surface concentrations of all species can be systematically affected by the seasonality of the boundary layer (BL) height (Baars et al., 2008). The compression effect of the lower BL height in winter than in summer systematically increases the near-surface concentrations.

The details for the different species are discussed as follows:

1) $NO_2$ and $SO_2$

$NO_2$ (rapidly formed from $NO_x$ after its emission) and $SO_2$ originate mostly from direct emissions. It is assumed that about 94% of total $NO_x$ emission in the Wuxi region is emitted from the power plants, industrial fuel combustions and vehicles (Huang et al., 2011), which emit similar amounts in different seasons. Thus the seasonal variation of the MAX-DOAS results cannot be explained by the variation of the $NO_x$ emissions. However, the $SO_2$ emissions might vary by about 20% due to the seasonal use of boilers for domestic heating (Huang et al., 2011). Because of the short lifetime of $NO_x$ under urban pollution (about some hours, e.g. Beirle et al., 2011 and Liu et al., 2015), most NOx should be from local emissions (Liu et al., 2015), and NOx long-range transport could be negligible in Wuxi. It needs to be noted that because of the longer life time of NOx in winter (Schaub et al., 2015) than in summer, transport of $NO_2$ from a nearby pollution area in winter might play a role on the seasonality of $NO_2$. Due to the large range of $SO_2$ residence time (from less than one hour to 2 weeks and longer in winter than in summer, e.g. von Glasow et al., 2009; Lee et al., 2011; Beirle et al., 2014), transport from the highly polluted regions in the east and north likely play a role, especially in winter. Here it is interesting to note that indications for long range transport of $SO_2$ are also found in the elevated $SO_2$ profiles in winter as shown in Fig 25b. Because of the strong seasonal variation of the $SO_2$ emissions due to domestic heating in the North (Wang et al., 2014a), long range transport from these regions could strongly impact the $SO_2$ amount in Wuxi in winter, thus contributing to the seasonality. In conclusion, the seasonality of $NO_2$ can be mostly attributed to the removal mechanisms due to the OH radical, which has a minimum in winter and maximum in summer (Stavrakou et al., 2013). The same removal mechanism could be partly responsible for $SO_2$ seasonality (Lee et al., 2011). Additional heterogeneous reactions (Oppenheimer et al., 1998) might also play a role. Since we find a high correlation between the $NO_2$ and $SO_2$ VCDs and near surface VMRs (see Fig. S24 of the supplement), we conclude that also for $SO_2$ the seasonality of the removal mechanism is the most important factor controlling the seasonality of the $SO_2$ VCDs and near-surface VMRs.

2) HCHO

HCHO originates mainly from the oxidation degradation of many VOCs by the OH radical. But because the OH radical also plays a role in the removal mechanism of HCHO, the seasonal variation of the OH radical level contributes to the seasonality of HCHO in a complex way. Apart from the ubiquitous background levels of HCHO from the methane oxidation, emissions of non-methane VOCs (NMVOCs) (including HCHO) from biogenic sources, biomass burning and anthropogenic sources control local HCHO concentrations. Therefore, in addition to the seasonality of OH, also the seasonal variations of the VOC emissions should be important factors for the HCHO seasonality. Firstly stronger biogenic emissions are expected in the





growing period, namely from spring to autumn. Based on a study in Beijing (Xie et al., 2008) a relative contribution of biogenic emissions to the total VOC levels is estimated at about 13%. Secondly, biomass burning events frequently occur in May and June (Cheng et al., 2014) in the Wuxi region. Thirdly anthropogenic emissions contribute a lot to the VOCs amounts. However the dominating sources, such as non-combustion industrial processes and vehicles (Huang et al., 2011), do not show an obvious seasonality. Thus, their effect on the HCHO seasonality can be probably ignored. Fourth, biogenic primary emissions of HCHO could be another factor contributing to the HCHO seasonality due to its significant differences between in summer and winter (Chen et al., 2014).

3) Aerosols

The local aerosol sources, including primary aerosol emissions and secondary aerosol formations, and transport of aerosols can in principle both contribute to the local aerosol amount. The contribution of transported aerosols has an obvious seasonality: In May and June, the transport from biomass burning might contribute to up to 37% of the $PM_{2.5}$ amount based on a case study in summer 2011 (Cheng et al., 2014). In spring and autumn dust storms from Mongolia can reach Wuxi (Liu et al., 2012; Fu et al., 2014b and Li et al., 2014). The polluted air from the eastern area (for example, Shanghai) and northern area (for example Jing–Jin–Ji region) (Jiang et al., 2015) could also move to Wuxi under appropriate meteorological conditions (Liu et al., 2012). Haze events frequently occur in autumn and winter (Fu et al., 2014a).

The inter-annual trends of TGs and aerosols are presented in Fig. 24. Because of missing observations in some months and inner-annual variations of abundances of the species, only data in May to November are used. $SO_2$ shows a clear decreasing trend from 2011 to 2014. However $NO_2$, HCHO and aerosols almost maintain constant amounts.

The monthly mean profiles of $NO_2$, $SO_2$ and HCHO (under clear and cloudy sky conditions except thick clouds and fog) and aerosols (only under clear sky conditions) (screened by the scheme in Table 3) are presented in Fig. 25. The monthly mean TG profiles under clear sky conditions (see Fig. S25 of the supplement) are almost identical to those under various sky conditions except fog and thick clouds in Fig. 25. During all seasons, $NO_2$ shows an exponentially decreasing profile (see Fig. 25a). On average the $NO_2$ VMR at 0.5 km is about half of the near-surface VMR and it rapidly decreases above 0.5km to about 2 ppb at 1.5 km. Aircraft measurements of $NO_x$ in October 2007 in the Yangtze River Delta region by Geng et al. (2009) presented similar vertical profiles. The profile shape of $NO_2$ can be mostly attributed to its near-surface emission sources and short life time.

The $SO_2$ layer is found at a higher altitude compared to $NO_2$ (see Fig. 25b). A more box-like shape up to the altitude of about 0.7km to 1km is found in autumn and winter when the $SO_2$ load is large and also long-range transport might effectively contribute to the $SO_2$ amounts in Wuxi. In contrast, for the rather small $SO_2$ loads in summer, an exponential profile shape is found. Similar profile shapes are also obtained from aircraft measurements during September to October of 2007 over Wuxi (Xue et al., 2009). One interesting finding is the lofted $SO_2$ layer at around 0.7 km in February and March 2012, which is probably related to long distance transport from a heavily polluted region. This interpretation is supported by the dominating wind direction (coming from the nearby polluted area around Shanghai) in March 2012 (see Fig. S26 of the supplement) compared to other years.



In all seasons, the HCHO profile shape consists of three parts (see Fig. 25c): a decrease from the surface to about 0.3 km, an almost constant value from about 0.3 km to about 1.1 km, and a steep decrease above. The high values at the surface are probably caused by primary emissions and rapid formation from particular VOCs near the surface. Transport of longer lived VOCs to higher altitudes and subsequent destruction probably contributes to the increased values at up to about 1 km. While

other measurements of tropospheric profiles of HCHO are not available around Wuxi, it is still reasonable to compare our results with the aircraft measurements of HCHO over Bresso near Milano during summer of 2003 (Junkermann, 2009; Wagner et al., 2011) because both of the measurements took place in polluted urban regions. They found a layer height with high HCHO concentration values of up to 1km and the highest values were found normally close to the ground. This feature is consistent with our results in Wuxi.

Fig. 25d shows the aerosol profiles representing a box-like shape near the surface and an exponential decrease above 0.5 to 1 km. The box-like part in winter is systematically lower than in other seasons probably due to the lower BL in winter. Baars et al. (2008) reported such a seasonal dependence of the top height of the BL obtained by lidar observations in Germany over a one-year period. A similar seasonal dependence of the BL can be expected in Wuxi. From May to October the highest aerosol extinction is found at an elevated altitude of up to 0.7km, especially in 2014. This feature could indicate long

distance transports of aerosols, probably from biomass burning events.

### 3.3 Diurnal variations of $NO_2$, $SO_2$, HCHO and aerosols

Fig. 26 shows the seasonally averaged diurnal variations of TG VCDs and near-surface VMRs as well as AODs and near-surface AEs from 2011 to 2014. The morning and afternoon averaged profiles of aerosols and TGs are also shown in winter and summer, respectively, in Fig. S27 of the supplement. The diurnal variations can probably be attributed to the complex

interaction of the primary and secondary sources, depositions and atmospheric transport processes in the BL. The diurnal variation of the BL height (Baars et al., 2008) can systematically affect the diurnal patterns of near-surface VMRs and AEs, but has almost no impact on the TG VCDs and AOD.

As seen in Fig. 26a, the seasonality of the diurnal variation of the $NO_2$ VCDs is quite similar to the MAX-DOAS observations in Beijing (Ma et al., 2013). They conclude that the phenomenon is probably caused by the complex interplay

of the emission, chemistry and transport, with generally higher emission rates and a longer $NO_2$ lifetime in winter. In Fig. 26b, the $SO_2$ VCD shows almost constant values during the whole day in summer (with a slight decrease in the afternoon). In winter high values persist until 13:00 LT and then rapidly decrease. In autumn and spring the highest values occur around noon. The $SO_2$ variation mostly happens in the layer below 0.5 km (see Fig. S27 of the supplement). The variation features are different from the observations in Beijing (Wang et al., 2014a), probably caused by different sources, transport and life

time at the two locations. In Fig. 26c it is shown that the HCHO VCDs increase rapidly after sunrise with a faster increase in summer. HCHO has a stronger variation at the layer from 0.5km to 1km. This diurnal pattern is probably mainly related to the photochemical formation of HCHO and the VOCs emitted by vehicles and biogenic emissions (Kesselmeier and Staudt, 1999). In Fig. 26d similar relative diurnal variations of AODs and AE are found for the different seasons. The decrease of



AOD from sunrise to around 9:00 LT might be caused by the decrease of the relative humidity after sunrise as shown in Fig. 21c. The increase of AOD from about 9:00 LT to noon might be caused by the photochemical formation of second aerosol particles. The decrease of AOD in the afternoon might indicate a reduced formation reaction rate.

### 3.4 Weekly cycles of $NO_2$, $SO_2$, HCHO and aerosol extinction

In urban areas, anthropogenic sources often control the amounts of pollutants. Because human activities are usually strongest during the working days, weekly cycles of $NO_2$, $SO_2$, HCHO and aerosols can provide information on the contributions from natural and anthropogenic sources (Beirle et al., 2003 and Ma et al., 2013). As shown in Fig. 27, weekly cycles are found for $NO_2$ and $SO_2$. The relative differences of the VCDs and near-surface VMRs between the average working day level (from Monday to Friday) and the value on Sunday are 11% and 18% for $NO_2$, 13% and 11% for $SO_2$, respectively. For HCHO smaller weekly cycles (7% of VCD and 12% of near-surface VMR ) are found:. In contrast, no clear weekend reduction is found for aerosols. The negligible weekly cycle of aerosols is probably caused by the rather long life time of aerosols and the effect of long-range transport, e.g. from biomass burning and dust. Fig. S28 of the supplement shows that the diurnal variations of the three TGs are almost the same on different days of a week indicating similar sources during the working days and weekends.

### 3.5 Source analysis of the pollutants

#### 3.5.1 Relation between the precursors and aerosols

Huang et al. (2014) showed that secondary aerosols including organic and inorganic aerosols (nitrates and sulfates) contribute to about 74% of the $PM_{2.5}$ mass collected during high pollution events in January 2013 at the urban site of Shanghai. The aerosol in Wuxi close to Shanghai is expected to have similar properties. $NO_x$ ($NO_2$ and NO) and $SO_2$ are the precursors of secondary inorganic aerosols through their conversion into nitrates and sulfates, respectively. HCHO can be used as a proxy for the local amount of VOCs, which are precursors of secondary organic aerosols (Claeys et al., 2004). We have investigated the relationship between aerosols and their precursors through a correlation study as in Lu et al. (2010), Veefkind et al. (2011) and Wang et al. (2014a). Table 5 lists the correlation coefficients between the TG VCDs and AODs as well as the TG VMRs and AEs near the surface. The correlations of near-surface values are always higher than those of the column densities. This finding could be probably explained by the effect of long-range transport, which typically occurs at elevated layers. For long-range transport, the effect of different atmospheric lifetimes is especially large probably leading to weaker correlations between the aerosol and its precursors. In contrast, close to the surface, local emissions dominate the concentrations of TG and aerosols and the effect of different lifetimes is negligible.

In general, correlations in spring are worst probably due to the transport of dust and biomass burning aerosols. The correlations between aerosols and HCHO are higher in winter than in summer. This finding may be explained by the fact that anthropogenic emissions dominate the (primary and secondary) sources of HCHO and aerosols simultaneously in winter.



Meanwhile the correlations between aerosols and HCHO are higher than those between aerosols and $NO_2$ or $SO_2$ in winter and autumn. This finding can be possible explained by the fact that both HCHO and aerosols are dominated by secondary sources, while $NO_2$ and $SO_2$ are mostly from primary emissions in this region.

### 3.5.2 Wind dependence of the pollutants

The MAX-DOAS station is located on the boundary of the urban and suburban areas as shown in Fig. 1b. Several iron factories, cement factories, and petroleum industries are operated in the south-west industrial area. The industrial activities and vehicle operations in the industrial area lead to significant emissions of $NO_2$, $SO_2$, VOCs as well as aerosols (Huang et al., 2011). In the urban centre area, traffic, construction sites and other anthropogenic emissions emit significant amounts of $NO_2$, VOCs as well as particles. Some factories, such as an oil refinery, are located in the north-west of the urban centre,

emitting pollutants including $SO_2$ and VOCs. In addition, one power plant located at about 50km in the north and the Suzhou city in the south-east direction of the MAX-DOAS station might contribute to the observed pollutants in Wuxi depending on the meteorological condition.

We analysed the distributions of column densities and near-surface values of the TGs and aerosols for different wind directions in Fig. 28. In principle, the near surface pollutants are expected to be dominated by nearby emission sources,

while the column densities can be additionally affected by transport of pollutants from remote sources. Long-range transport can weaken the dependence of the column densities on the wind direction because of the complex trajectories the air masses might have followed. For all four species, the highest values are observed for south-westerly winds, especially for the near-surface pollutants. This finding implies that the industrial area emits large amounts of $NO_x$, $SO_2$, VOCs and aerosols. Fig. 28c shows that the HCHO southwest peak is only present in winter. This finding is probably caused by the fact that in winter

anthropogenic sources (of precursors and direct HCHO emissions) dominate the HCHO amounts, while in other seasons natural sources dominate the HCHO amounts. Another peak of $NO_2$ and $SO_2$ is found in the northwest, obviously in winter, indicating considerable emissions in the urban centre. Fig 28d shows a weaker dependence of AODs on the wind direction than the VCDs of the TGs, which probably indicates the stronger contribution of long-range transport to the local aerosol levels compared to the TGs. In addition for daily averaged wind speed of smaller than 1 m/s, the averaged TGs VCDs and

near-surface VMRs are higher than those for larger wind speeds (shown in Fig. 29a and b), indicating that dispersion of local emissions is more important than the transport from distant sources. For aerosols, a wind speed dependency is only observed for near-surface AEs, but not for AODs (see Fig. 29c), indicating the higher importance of transport for aerosols than for TGs.

### 4 Conclusions

The long-term characteristics of the spatial and temporal variation of $NO_2$, $SO_2$, HCHO and aerosols in Wuxi (part of the Yangtze River delta region) are characterized by automatic MAX-DOAS observations from May 2011 to Dec 2014. The



PriAM OE-based algorithm was applied to MAX-DOAS observations to acquire vertical profiles, VCDs (AODs) and near-surface VMRs (AEs) of TGs (aerosols) in the layer from the surface to an altitude of about 3 km.

The AODs and near-surface AEs and the VMRs of $NO_2$ and $SO_2$ from MAX-DOAS are compared with coincident data sets (for one year) obtained by a sun photometer at the AERONET Taihu station, a nearby visibility meter and a LP-DOAS,

respectively. In general good agreement was found: Under clear sky conditions, correlation coefficients of 0.56–0.91 for AODs, 0.31–0.71 for AEs, 0.42–0.64 for $NO_2$ VMRs and 0.68–0.81 for $SO_2$ VMRs as well as the low systematic bias of -0.16–0.029 for AODs, 0.05–0.19 $km^{-1}$ for AEs, -2.23–5.11 ppb for $NO_2$ VMRs and 1.8–6.1 ppb for $SO_2$ VMRs are found in different seasons.

Further comparisons were performed for different cloud conditions identified by the MAX-DOAS cloud classification

scheme (Wagner et al., 2014 and Wang et al., 2015). For most cloud conditions (except optically thick clouds and fog) similar agreement as for clear sky conditions is found for the results of near-surface TG VMRs and AEs. However, the AOD results are more strongly affected by clouds and we recommend to only retrieve near-surface AEs for cloudy observations. In the presence of fog and optically thick clouds, no meaningful profile inversions for TGs and aerosols are possible. Thus for further interpretations, we considered TG results and near-surface AEs for clear and cloudy sky conditions (except fog and

optically thick clouds), but AOD only for clear sky conditions.

In this study we also investigated two important aspects of the MAX-DOAS data analysis: For the first time the effect of the seasonality of temperature and pressure on the MAX-DOAS retrievals of aerosols was investigated. Such an effect is especially important for the measurements in Wuxi, because strong and systematic variations of temperature and pressure are regularly found. Accordingly the $O_4$ VCD changes systematically with seasons, which was in our study for the first time

explicitly taken into account for the aerosol profile retrieval. It was shown that without this correction, deviations of the AOD of up to 20% can occur.

Moreover, we systematically compared trace gas VCDs derived either by the so-called geometric approximation with those derived by integration of the derived vertical profiles. Such discrepancies were reported in previous studies. We could show that the difference between both methods can be clearly assigned to limitations of the geometric approximation. This error

becomes especially significant when the aerosol load is strong, which is the situation in most industrialised regions. Thus we conclude that in general the integration of the retrieved profiles is the more exact way to extract the tropospheric TG VCDs, and we used this method in this study.

A prominent seasonality of all TGs is found in agreement with many previous studies based on satellite and ground-based observations. $NO_2$ and $SO_2$ have maxima and minima in winter and summer, respectively, while HCHO has an opposite

seasonality. No pronounced seasonality of aerosols is found. From 2011 to 2014, only $SO_2$ shows a clear decreasing trend, while $NO_2$, HCHO and aerosol levels stay almost constant.

Different profile shapes are found for the different species: for $NO_2$ exponentially decreasing profiles with a scale height of about 0.6km are observed in different seasons. $SO_2$ profiles extend to slightly higher altitudes than $NO_2$, probably due to the



longer lifetime of $SO_2$. Especially in winter often elevated layers of enhanced $SO_2$ are found between about 0.7km and 1km (especially in early 2012), probably indicating the importance of long range transport of $SO_2$. HCHO reaches up to even higher altitudes (up to > 1 km) than $NO_2$ and $SO_2$, probably indicating the effect of the secondary formation from VOCs. However, typically the largest HCHO VMRs are still found near the surface (like for $NO_2$ and $SO_2$). The aerosol profiles

typically show constant values close to the surface (below about 0.5km), but decrease exponentially above that layer. Especially in winter often elevated layers (between 0.5km and 0.7km) are observed.

Different diurnal variations are found for the different species: For the $NO_2$ VCDs, depending on season, a decrease or increase is found during the day. For the $NO_2$ VMRs and $SO_2$ VCDs and VMRs, typically a slight decrease during the day is observed. The diurnal variations of HCHO and aerosols are more complex and show a pronounced maximum around noon in

summer indicating photochemical production. Systematic weekly cycles occur for $NO_2$ and $SO_2$ with the maximum values on Thursday or Friday and minimum values on Sunday indicating a large contribution of anthropogenic emissions. In contrast, the amplitudes of the weekly cycles for HCHO and aerosols are rather small.

We performed correlation analyses between the different TG results versus the aerosol results for individual seasons. For all TGs and seasons positive correlations (correlation coefficient between 0.12 and 0.65) were found with the highest

correlations in winter. In general the highest correlation is found for HCHO in winter probably indicating a similar secondary formation process for both species. In general, higher correlations are found for the near-surface products (VMRs versus AE) compared to the column products (VCDs versus AOD).

We found a clear wind direction dependence of TG and aerosols results, especially for the near-surface concentrations. The dependencies indicate that the largest sources of the observed pollutants in Wuxi are anthropogenic emissions from the

nearby industrial area (including traffic emissions). In addition the obvious lower TG results for high wind speed than for low wind speed indicate that the dispersion of local emissions is more important than the transport from distant sources. Interestingly, for HCHO, a considerable dependence on the wind direction is only observed in winter probably indicating significant VOC emissions from natural sources in the growing seasons.

The data sets of the TGs and aerosols are also valuable to validate tropospheric products from satellite observations and

chemical transport models. This study is in progress.

**Acknowledgements:** We thank the Wuxi CAS Photonics Co. Ltd for their contributions to operate the observations of the MAX-DOAS instrument, the long path DOAS instrument, the visibility meter and the weather station in Wuxi. We thank the Institute of Remote Sensing / Institute of Environmental Physics, University of Bremen, Bremen, Germany for their freely

accessible radiative transfer model SCIATRAN. We thank Belgian Institute for Space Aeronomy (BIRA-IASB), Brussels, Belgium for their freely accessible WINDOAS software and generating the mean map of $SO_2$ and HCHO tropospheric VCDs derived from OMI observations over eastern China. We thank Goddard Space Flight Center, NASA for their freely accessible archive of AERONET data. We thank Prof. Ma Ronghua in Nanjing Institute of Geography & Limnology Chinese Academy of Sciences for his effort to operate the Taihu AERONET station. We thank Royal Netherlands





Meteorological Institute for their freely accessible archive of OMI tropospheric NO$_2$ data. We thank Trissevgeni Stavrakou in BIRA-IASB, Hang Su and Yangfang Cheng in MPIC for their help to interpret the correlations between aerosols and the trace gases. This work was supported by Max Planck Society-Chinese Academy of Sciences Joint Doctoral Promotion Programme, and National Natural Science Foundation of China (Grant No.: 41275038 and 41530644) and Monitoring and Assessment of Regional air quality in China using space Observations, Project Of Long-term sino-european co-Operation (MarcoPolo), FP7 (Grant No: 606953).

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



**Tables**

Table 1 Setting used for the $O_4$, $NO_2$, $SO_2$ and HCHO DOAS analyses

| Parameter | Data source | Fitting interval | | | |
|---|---|---|---|---|---|
| | | $O_4$ | $NO_2$ | $SO_2$ | HCHO |
| Wavelength range | | 351-390nm | 351-390nm | 307.8-330nm | 324.6-359nm |
| $NO_2$ | Vandaele et al. (1998), 220 K, 294 K | × | × | ×(only 294 K) | ×(only 294 K), $I_0$-corrected[*] ($10^{17}$ molecules/cm$^2$) |
| $O_3$ | Bogumil et al., (2003), 223 K and 243 K | ×(only 223 K) | ×(only 223 K) | × | ×(only 223 K) $I_0$-corrected[*] ($10^{18}$ molecules/cm$^2$) |
| $O_4$ | Thalman and Volkamer (2013), 293 K | × | × | × | × |
| $SO_2$ | Bogumil et al. (2003), 293 K | | | × | × |
| HCHO | Meller and Moortgat (2000), 293 K | × | × | × | × |
| Ring | | × | × | × | × |
| | Two Ring spectra calculated with DOASIS (Kraus, 2006; Wagner et al., 2009) | | | | |
| Polynomial degree | | 3 | 3 | 5 | 5 |
| Intensity offset | | constant | constant | constant | constant |

* solar $I_0$ correction, Aliwell et al., 2002





Table 2 Different filters and corresponding thresholds applied to the retrieved SCDs. Also the corresponding fractions of screened data are shown. (SZA: solar zenith angle; RIO: relative intensity offset; RMS: root mean square of the spectral residual)

| O4 and NO$_2$ | | SO$_2$ | | HCHO | |
|---|---|---|---|---|---|
| filter | percentage | filter | percentage | filter | percentage |
| SZA $< 75°$ | 6.2% | SZA $< 75°$ | 5.8% | SZA $< 75°$ | 6.1% |
| RIO $< 0.01$ | 5.6% | RIO $< 0.01$ | 1.1% | RIO $< 0.01$ | 7.1% |
| RMS $< 0.003$ | 0.3% | RMS $< 0.01$ | 0.2% | RMS $< 0.003$ | 0.2% |

5    Table 3 Filter scheme of aerosol and trace gas results derived from MAX-DOAS observations. Filled circles (●): use of measurement is recommended; Open circles (○): use of measurement is not recommended.

| | AOD | Aerosol extinction near surface | Profile of aerosol extinction | VCD | VMR near surface | Profile of VMRs |
|---|---|---|---|---|---|---|
| Low aerosols | ● | ● | ● | ● | ● | ● |
| High aerosols | ● | ● | ● | ● | ● | ● |
| Cloud holes | ○ | ● | ○ | ● | ● | ● |
| Broken clouds | ○ | ● | ○ | ● | ● | ● |
| Continuous clouds | ○ | ● | ○ | ● | ● | ● |
| fog | ○ | ○ | ○ | ○ | ○ | ○ |
| Thick clouds | ○ | ○ | ○ | ○ | ○ | ○ |





Table 4 Averaged error budget (in %) of the retrieved TG VCDs and AOD, and near-surface (0–200 m) TG VMRs and AE. The total uncertainty is calculated by adding the different error terms in Gaussian error propagation.

| | 0-200 m | | | | VCD or AOD | | | |
|---|---|---|---|---|---|---|---|---|
| | AE | $NO_2$ | $SO_2$ | HCHO | AOD | $NO_2$ | $SO_2$ | HCHO |
| Smoothing and noise error | 10 | 12 | 19 | 50 | 6 | 17 | 25 | 50 |
| Algorithm error | 4 | 3 | 4 | 4 | 8 | 11 | 10 | 11 |
| Cross section error | 5 | 3 | 5 | 9 | 5 | 3 | 5 | 9 |
| Related to temperature dependence of cross section | 10 | 2 | 3 | 6 | 10 | 2 | 3 | 6 |
| Related to the aerosol retrieval (only for trace gases) | - | 16 | 16 | 16 | - | 15 | 15 | 15 |
| Total | 16 | 21 | 26 | 54 | 15 | 25 | 31 | 54 |

Table 5 Correlation coefficients between hourly averaged trace gas VCDs and AODs (for clear sky conditions) as well as between VMRs and aerosol extinction near the surface (for clear and cloudy conditions except thick clouds and fog). The numbers of the data point use for the analysis are given for each season.

| | winter | | spring | | summer | | autumn | |
|---|---|---|---|---|---|---|---|---|
| | column | surface | column | surface | column | surface | column | surface |
| Number of observations | 375 | 525 | 1339 | 1739 | 1308 | 1830 | 1142 | 1676 |
| $NO_2$ | 0.51 | 0.69 | 0.37 | 0.58 | 0.48 | 0.63 | 0.44 | 0.65 |
| $SO_2$ | 0.52 | 0.69 | 0.45 | 0.62 | 0.45 | 0.62 | 0.44 | 0.66 |
| HCHO | 0.77 | 0.81 | 0.51 | 0.62 | 0.35 | 0.62 | 0.57 | 0.69 |

(a)     (b)     (c)

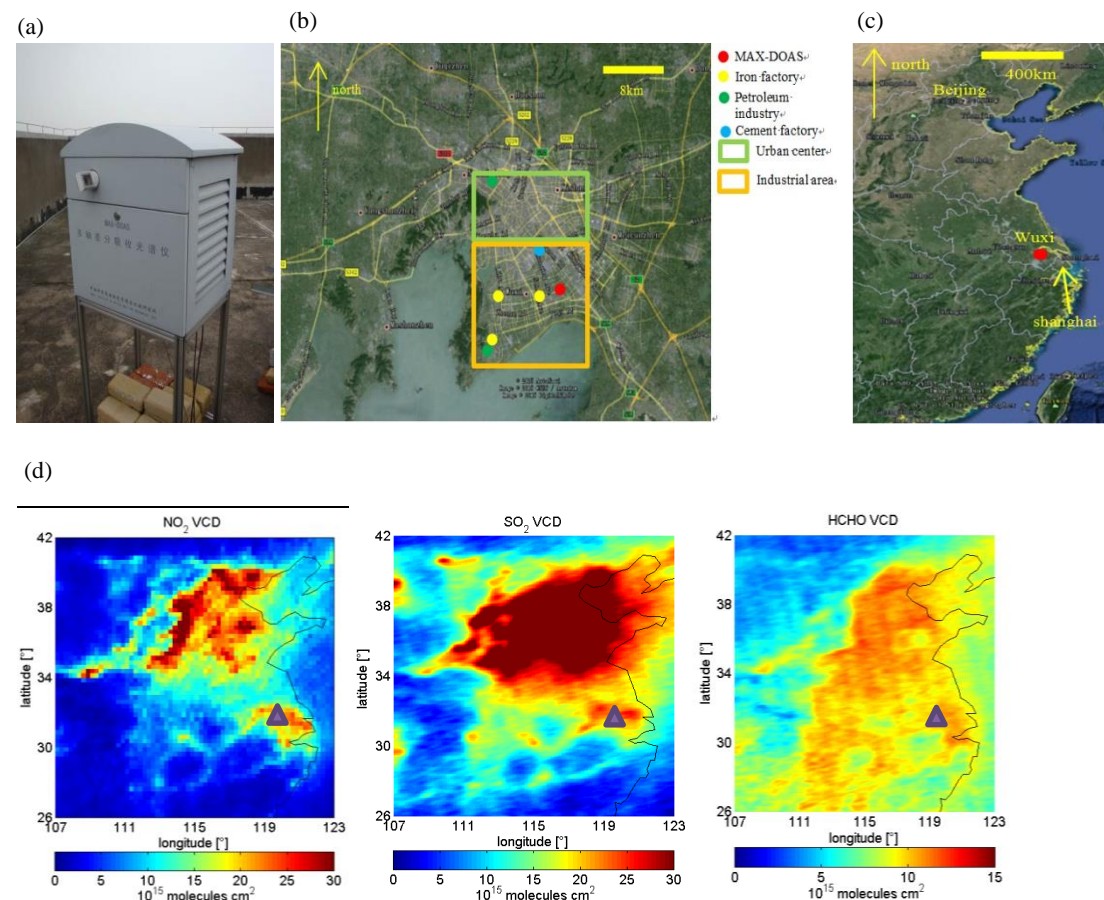

(d)

**Figure 1: The MAX-DOAS instrument (a) (also the long path DOAS and the visibility meter) is operated at the location marked by the red dot in subfigure (b) in Wuxi city (c). In subfigure (b), the dots with different colours indicate the positions of different types of emission sources; the green and orange blocks indicate the urban centre and industrial area, respectively. The mean maps of tropospheric VCDs of NO$_2$ (from DOMINO version 2), SO$_2$ (from BIRA, Theys et al., 2015) and HCHO (from BIRA, I. De Smedt et al., 2015) derived from OMI observations over eastern China in the period from 2011 to 2014 are shown in subfigure (d), in which the triangle flag indicates the location of Wuxi.**



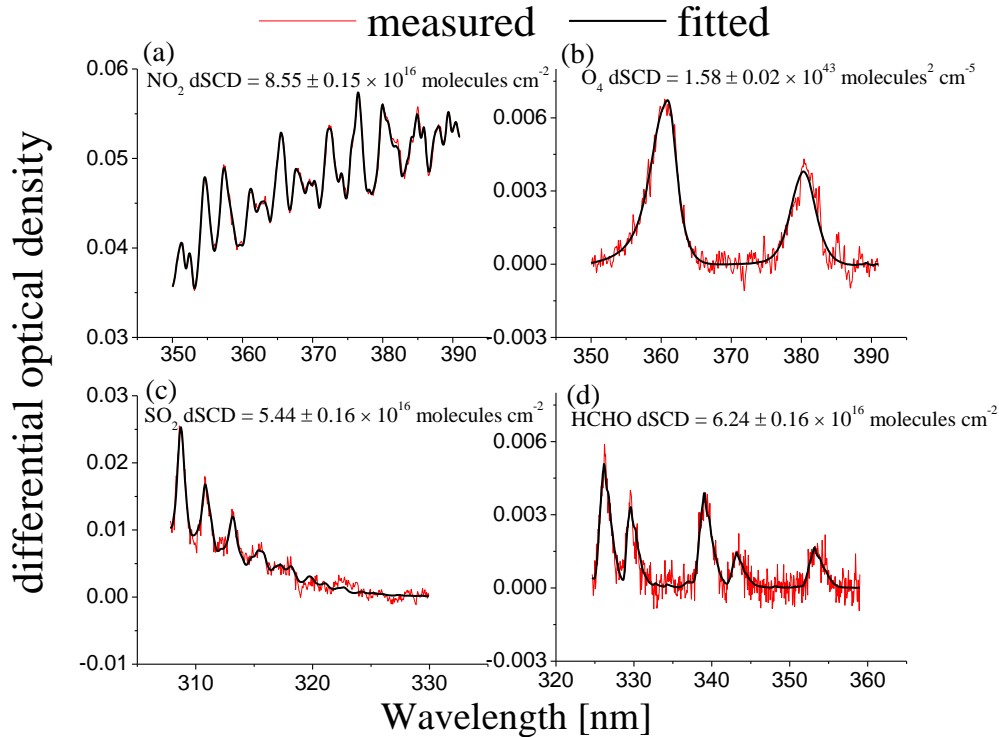

**Figure 2: Examples of typical DOAS fits of NO$_2$ (a), O$_4$ (b) and SO$_2$ (c) at 11:37 on 1 December 2011 as well as HCHO at 11:34 on 12 July 2012. The fitted dSCDs of NO$_2$, O$_4$, SO$_2$ and HCHO are given in the corresponding subfigures. The black and red curves indicate the fitted absorption structures and the derived absorption structures from the measured spectra, respectively.**

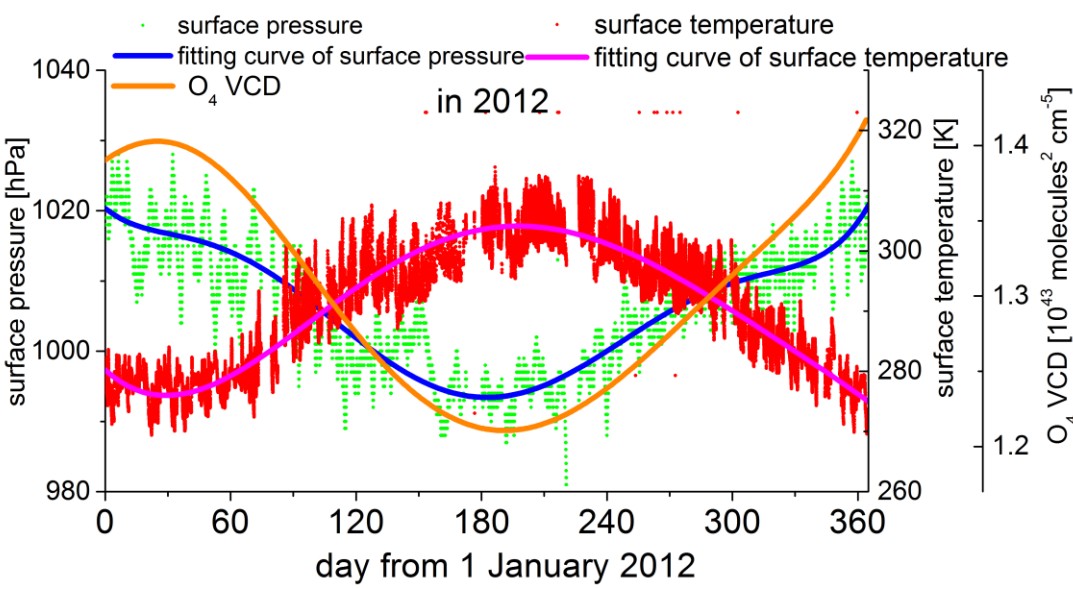

**Figure 3: Annual variation of surface temperature, surface pressure as well as fitted 6$^{th}$ order polynomials in 2012. Also the O$_4$ VCDs calculated based on the fitted curves of the measured annual variations of surface temperature and pressure in 2012 is shown (similar results are found for other years).**



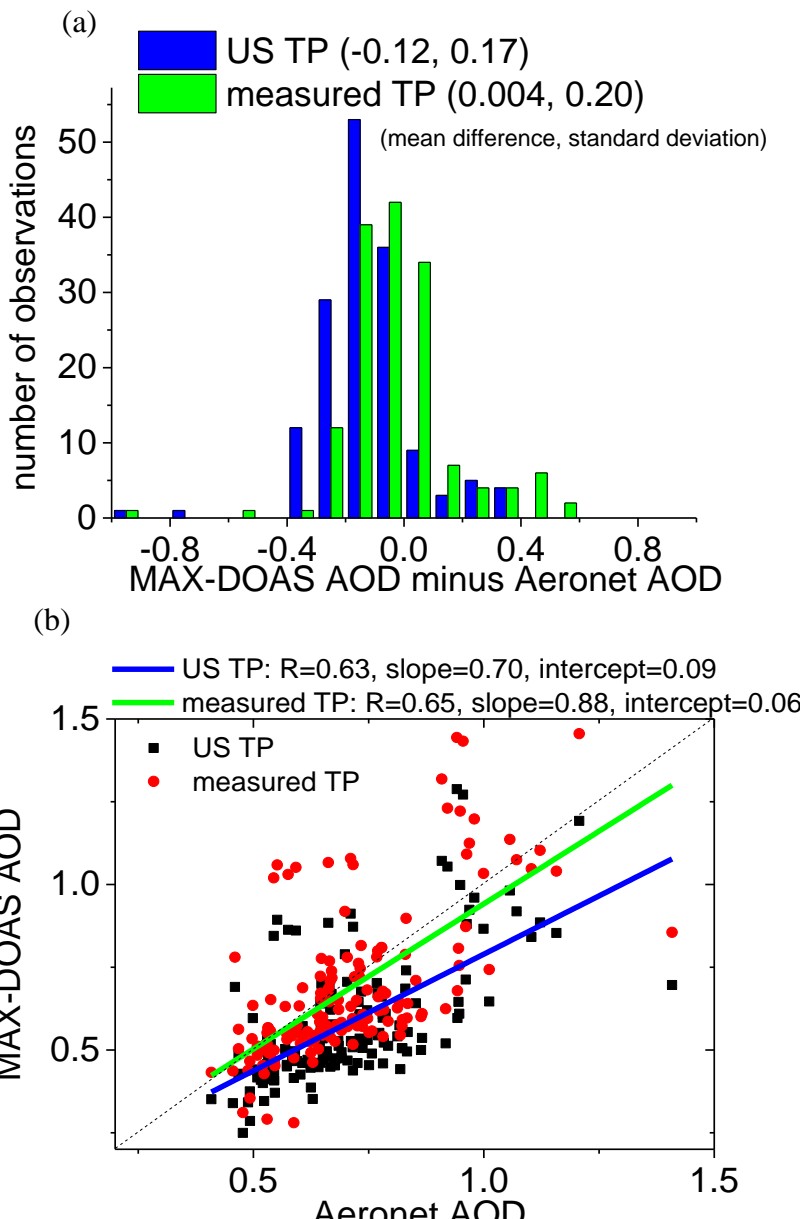

**Figure 4: Dependence of retrieved AOD on temperature and pressure (TP)for clear sky conditions. (a): Frequency distribution of the differences of the AODs derived from MAX-DOAS and AERONET for January 2012. The MAX-DOAS results from the retrieval using either the US standard summer TP profiles or the explicit TP from local measurements are indicated by blue and green colours, respectively. The mean difference and standard deviation are shown in brackets. (b): The AODs retrieved from MAX-DOAS observations (using either the US standard summer TP profile or the explicit TP from measurements) are plotted against those from the Taihu AERONET station. The results of the linear regression are shown on top of the diagram.**





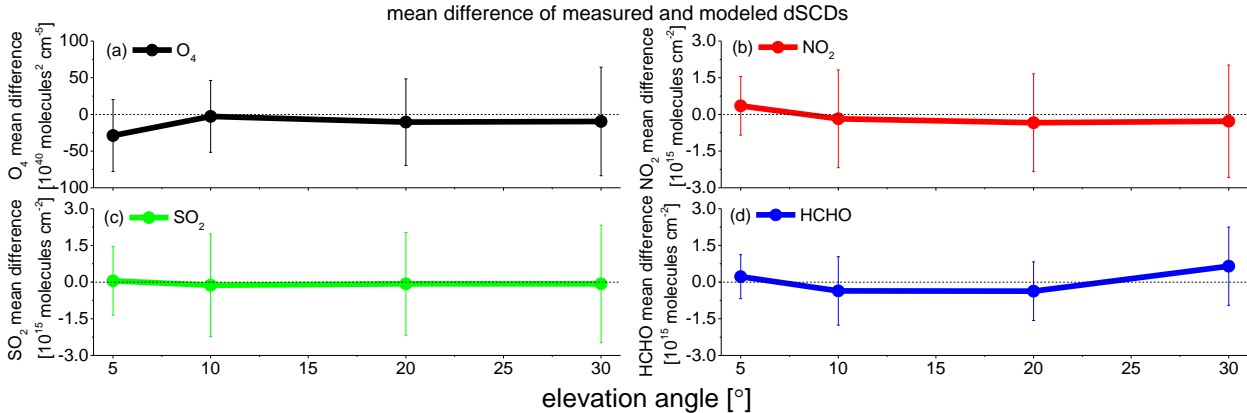

**Figure 5: Mean differences and the standard deviations (error bars) between the measured and modelled dSCDs of $O_4$ (a), $NO_2$ (b), $SO_2$ (c) and HCHO (d) for clear sky conditions with low aerosols plotted against the elevation angles of MAX-DOAS measurements.**

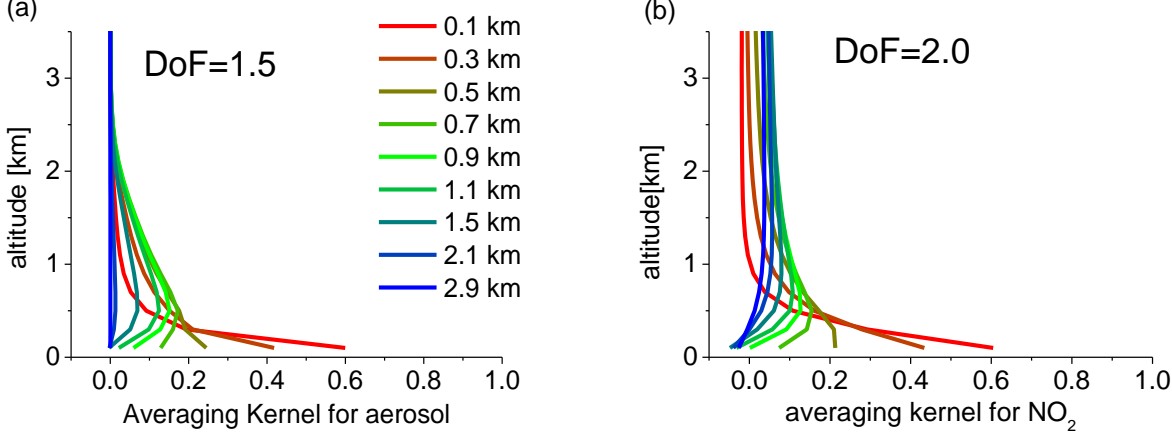

**Figure 6: Averaged averaging kernels of aerosol (a) and $NO_2$ (b) retrievals for all the MAX-DOAS measurements for clear sky conditions with low aerosols. DoF is the degree of freedom related to the averaging kernel.**



**Figure 7: The averaged profiles retrieved from the measurements in the whole period and in different season for clear sky**
5   **conditions with low aerosols: (a) aerosol extinction, (b) NO₂ VMR, (c) SO₂ VMR and (d) HCHO VMR. Also shown are the respective a-priori profiles.**



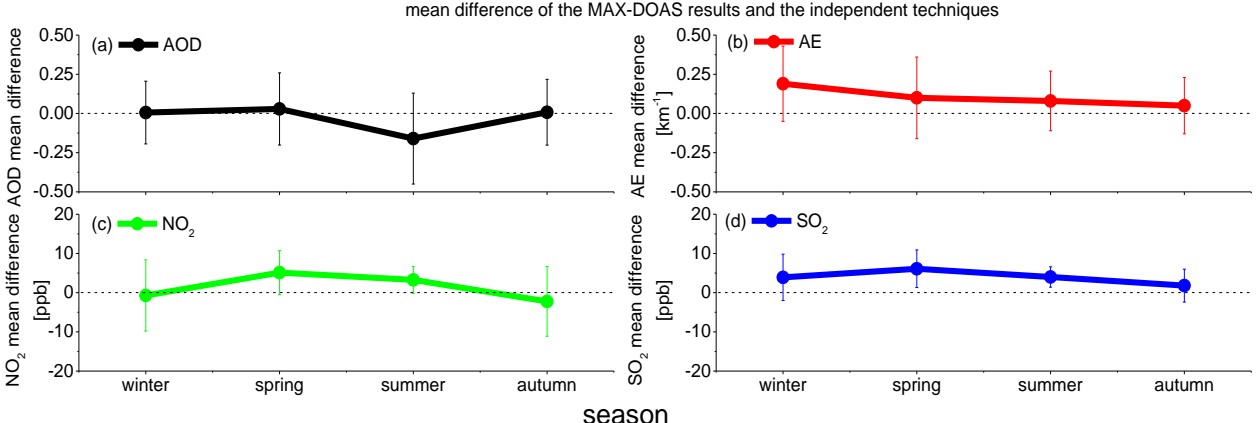

**Figure 8: Seasonally mean differences and standard deviations (shown as the error bar) between MAX-DOAS results and independent techniques for different seasons for clear sky conditions with low aerosols. Different colours denote AOD (a) (compared with the Taihu AERONET level 1.5 data sets), AE (b) (compared with the nearby visibility meter) and NO$_2$ (c) and SO$_2$ (d) (compared with the nearby long path DOAS instrument).**

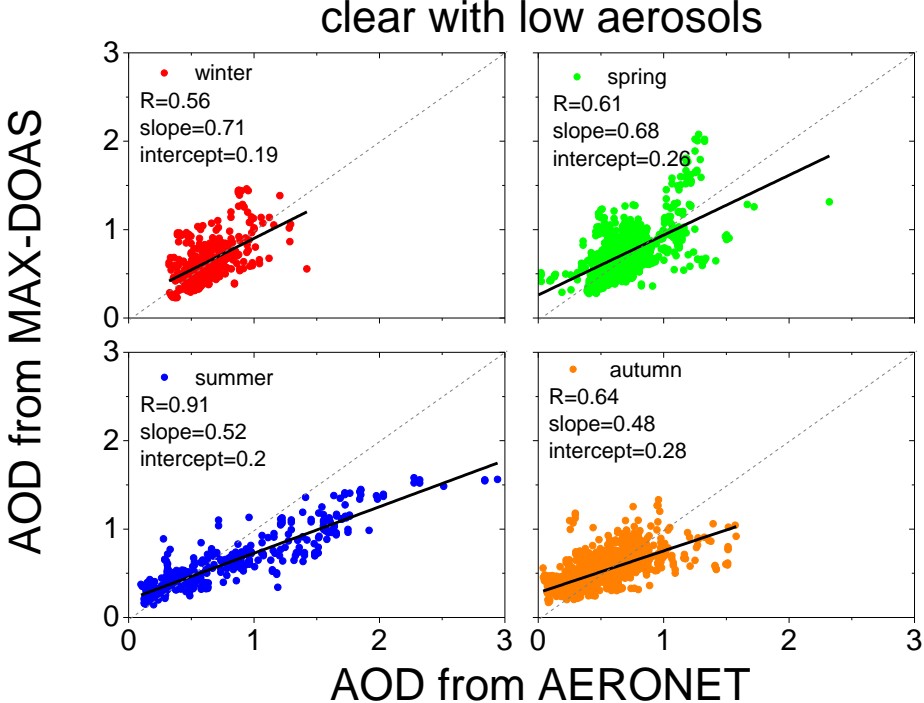

**Figure 9:, Scatter plots of the AODs derived from MAX-DOAS versus those from the Taihu AERONET station (level 1.5) in different seasons for clear sky conditions with low aerosols. Results of the linear regression are shown in the individual subfigures.**





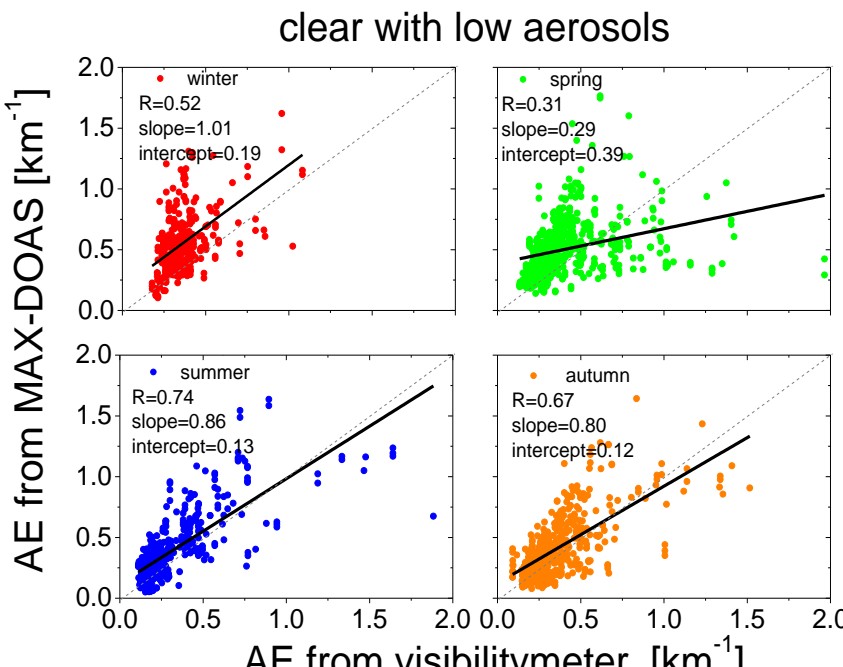

**Figure 10: Same as Fig. 9 but for the comparison of the near surface aerosol extinction derived from MAX-DOAS and the visibility meter**

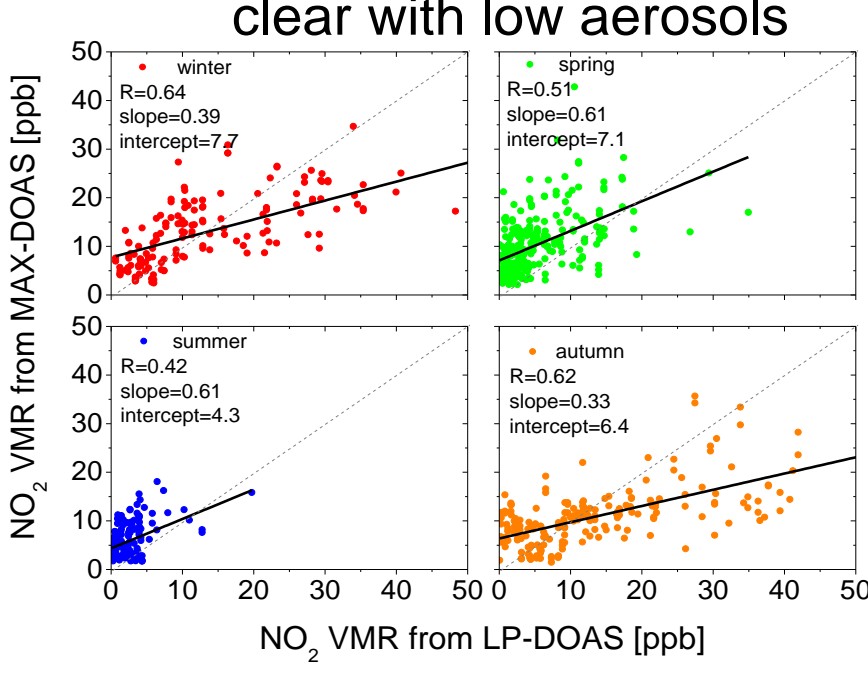

5   **Figure 11: Same as Fig. 9 but for comparison of the near surface NO₂ mixing ratios derived from MAX-DOAS and LP-DOAS.**





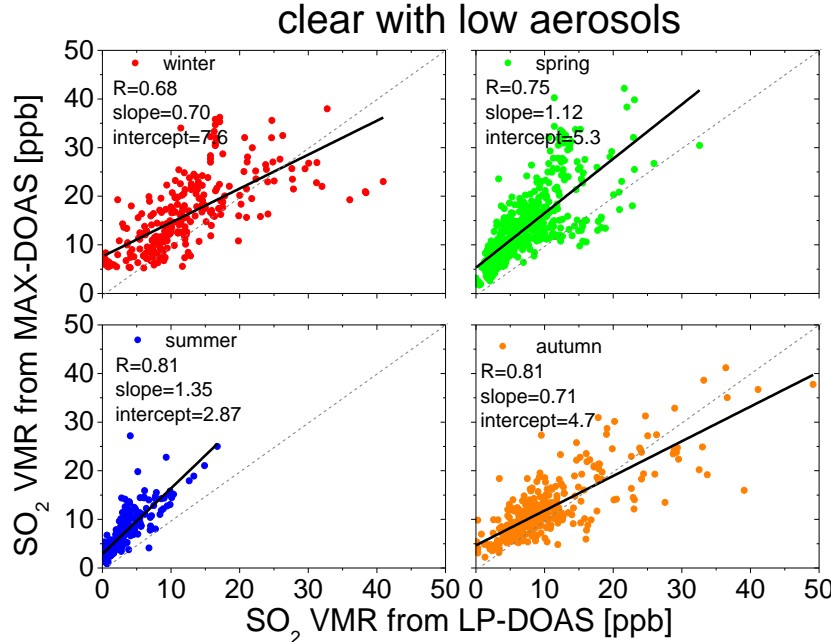

**Figure 12: Same as Fig. 9 but for comparison of the near surface SO₂ mixing ratios derived from MAX-DOAS and LP-DOAS.**

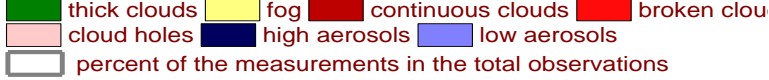

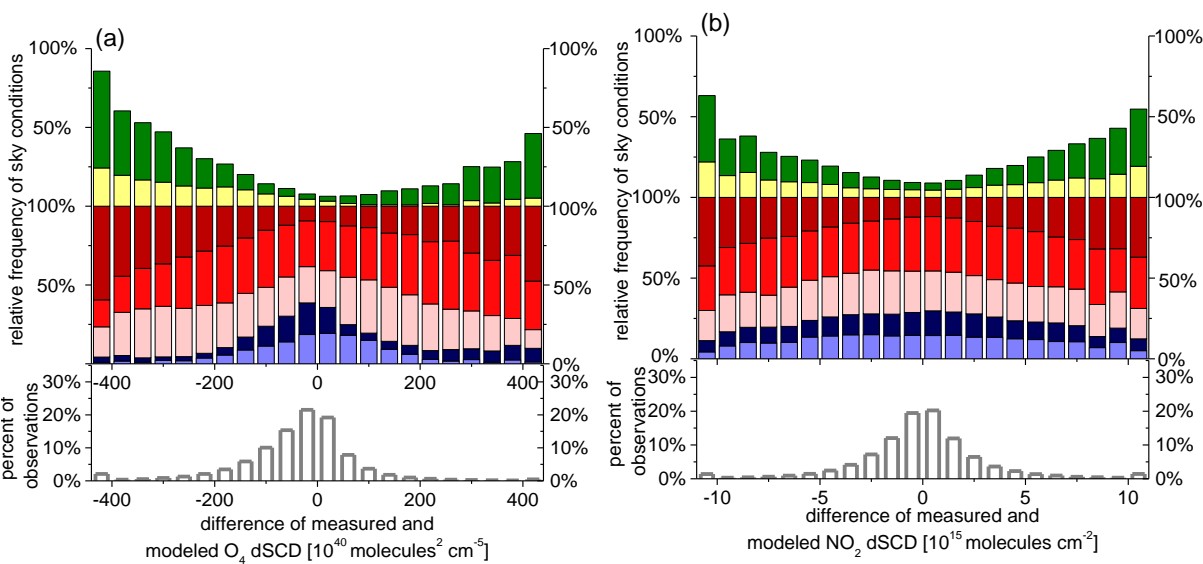



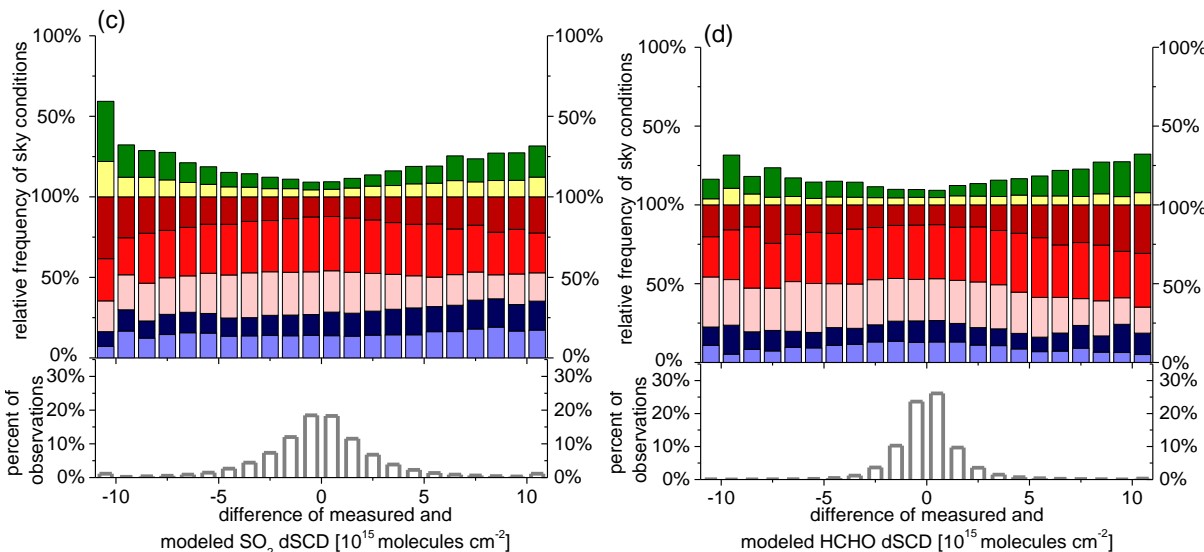

**Figure 13: Histograms of the differences between the measured and modelled dSCDs of $O_4$ (a), $NO_2$ (b), $SO_2$ (c) and HCHO (d) for all elevation angles. The colour bars show the relative frequencies of the different sky conditions for each bin (top). The grey hollow bars (bottom) represent the relative frequencies of the number of measurements compared to the total number of observations.**

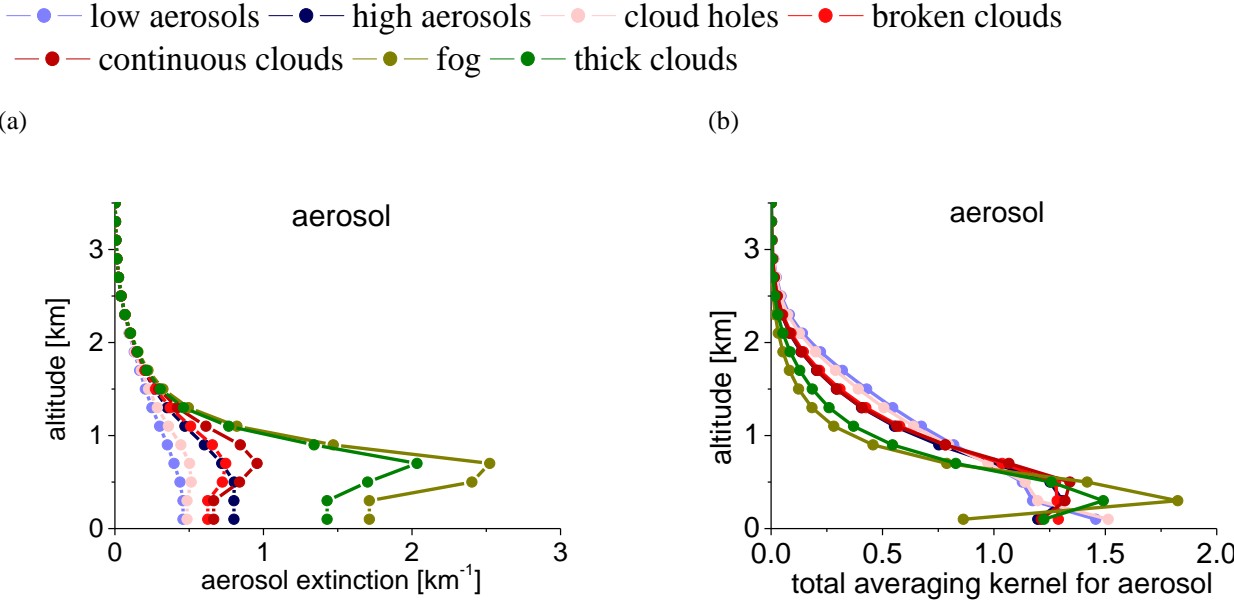





(c)

(d)

(e)

(f)

5   (g)

(h)



**Figure 14: Mean profiles of aerosol extinctions (a), NO$_2$ VMRs (c), SO$_2$ VMRs (e) and HCHO VMRs (g) from all MAX-DOAS observations under individual sky conditions; the subfigures (b), (d), (f), (h) show the total averaging kernels of the four species under individual sky conditions.**

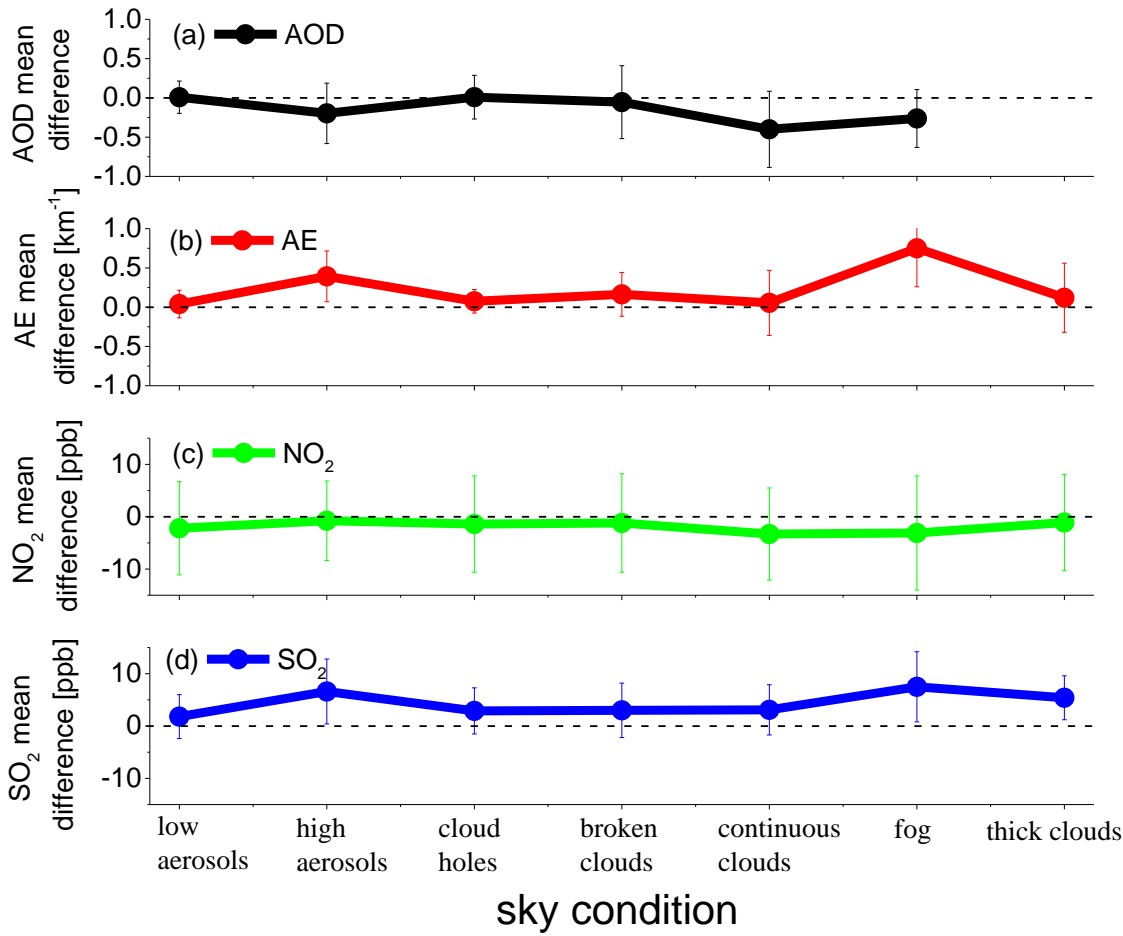

5    **Figure 15: Mean differences and the standard deviation (shown as the error bar) between MAX-DOAS results and independent techniques for different sky conditions. Different colours denote the values of AOD (compared with Taihu AERONET level 1.5 data sets), AE (compared with the visibility meter located nearby) and NO$_2$ and SO$_2$ (compared with the close long path DOAS instrument).**





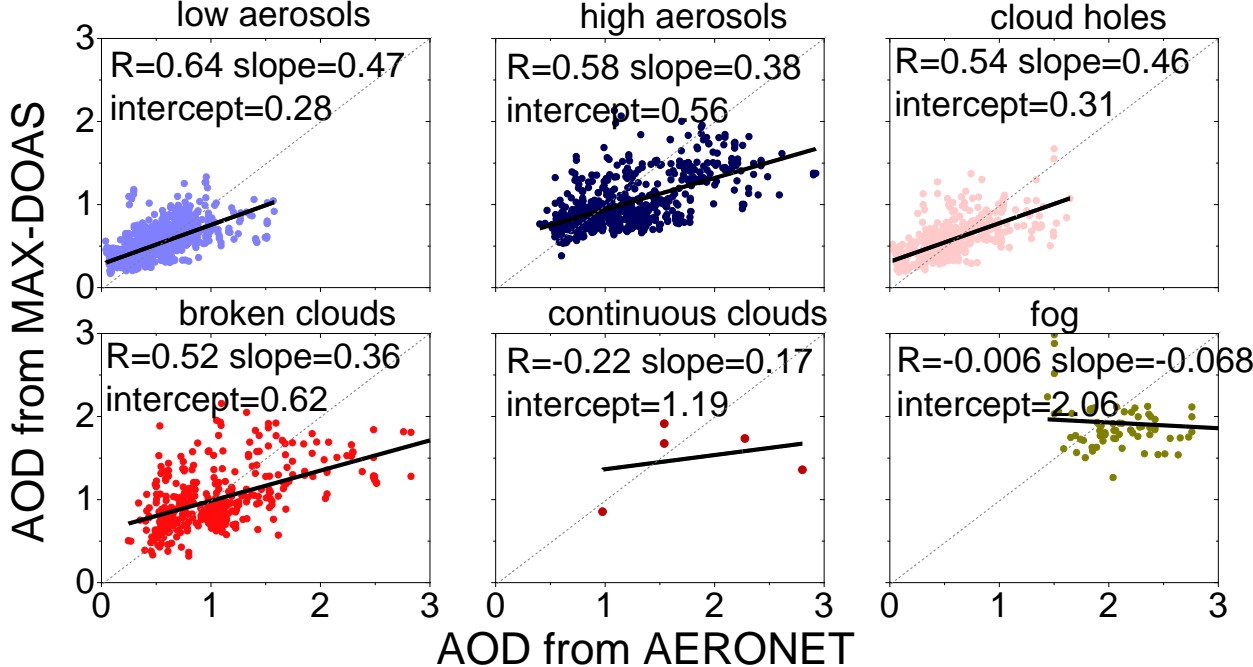

**Figure 16: AODs derived from MAX-DOAS measurements in autumn are plotted against those from AERONET for different sky conditions. The linear regression parameters are shown in each subfigure. Note that no AERONET level 1.5 AOD data is available for thick clouds conditions because of the AERONET cloud screening scheme.**

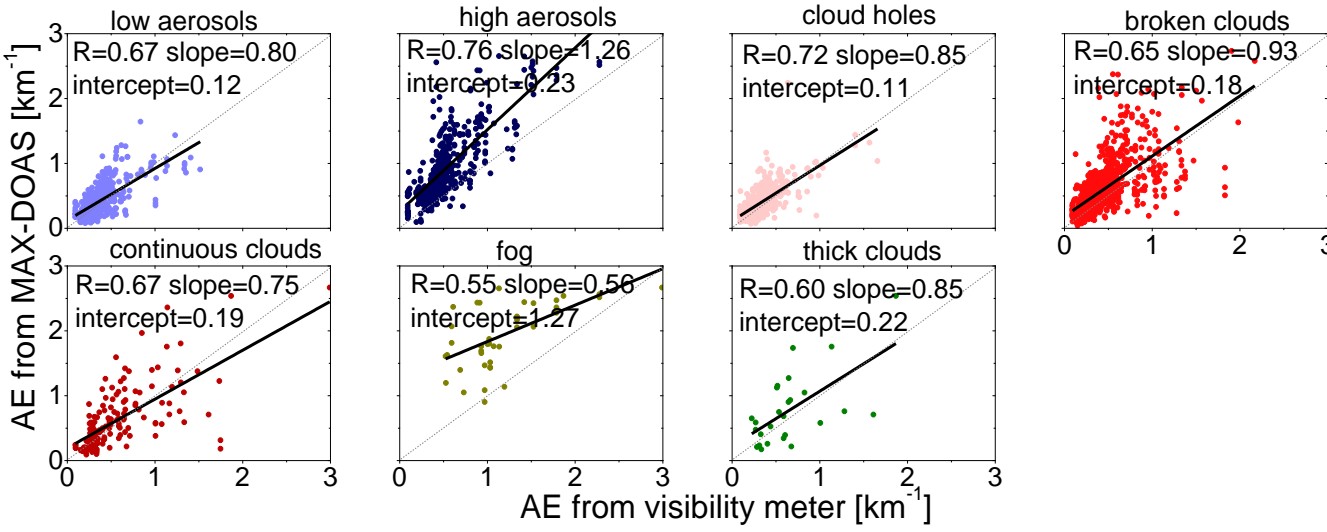

**Figure 17: Same as Fig. 16 but for the comparisons of near surface aerosol extinctions with the visibility meter.**





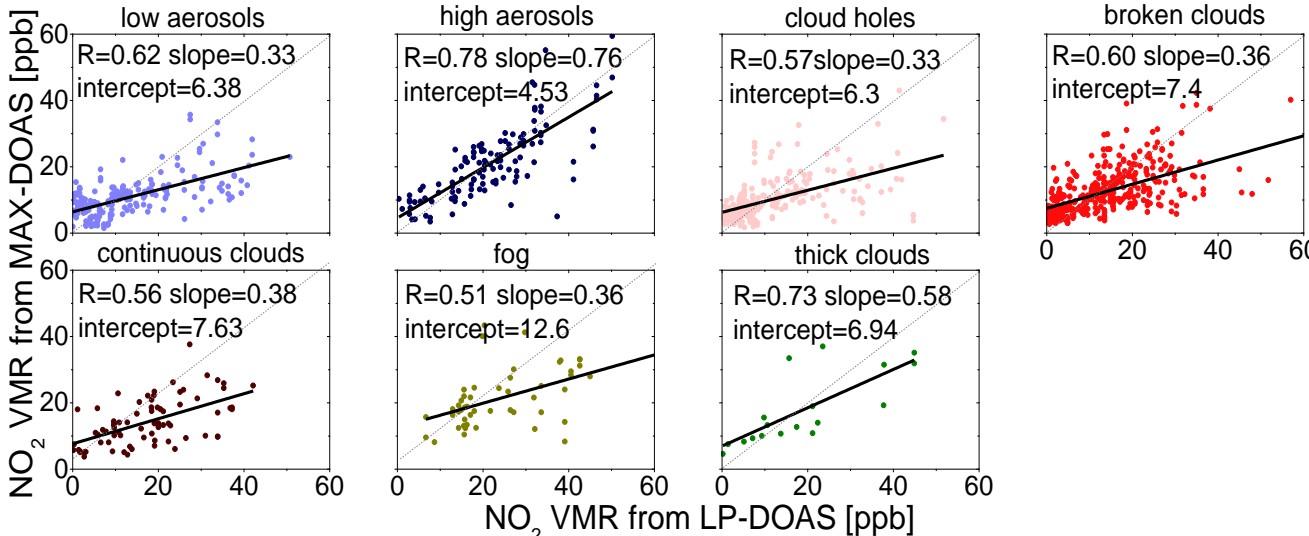

**Figure 18: Same as Fig. 16 but for the comparisons of near surface NO₂ VMRs with the LP-DOAS.**

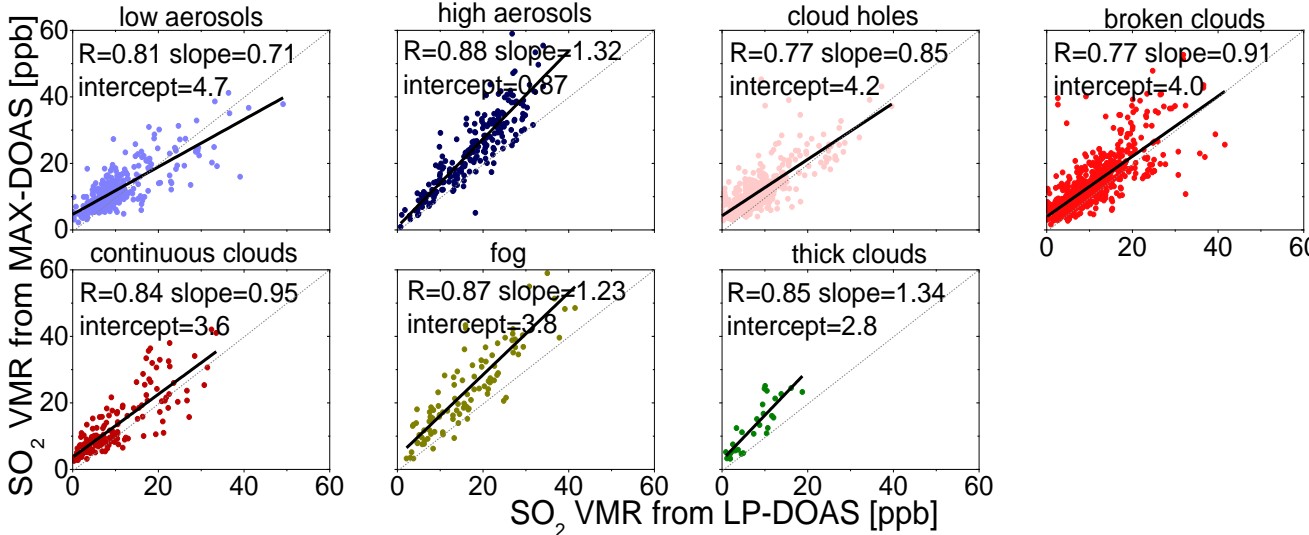

**Figure 19: Same as Fig. 16 but for the comparisons of near surface SO₂ VMRs with the LP-DOAS.**



**Figure 20: Relative differences of the tropospheric NO$_2$ (top row), SO$_2$ (middle row) and HCHO (bottom row) VCDs derived by the geometric approximation and from the profile inversion (Diff$_{total}$, black dots) as function of the relative azimuth angle for elevation angles of 20 ° (left) and 30 ° (right). Also the differences caused by the errors of the profile retrieval ($Diff_{inversion}$, red dots) and of the geometric approximation ($Diff_{geometry}$, blue dots) are shown (see text).**





Figure 21: Seasonally mean diurnal variations (2011 to 2014) of ambient temperature (a), wind speed (b) and relative humidity (c) obtained from the observations of the weather station nearby the MAX-DOAS instrument.





(a)

(b)

(c)

(d)

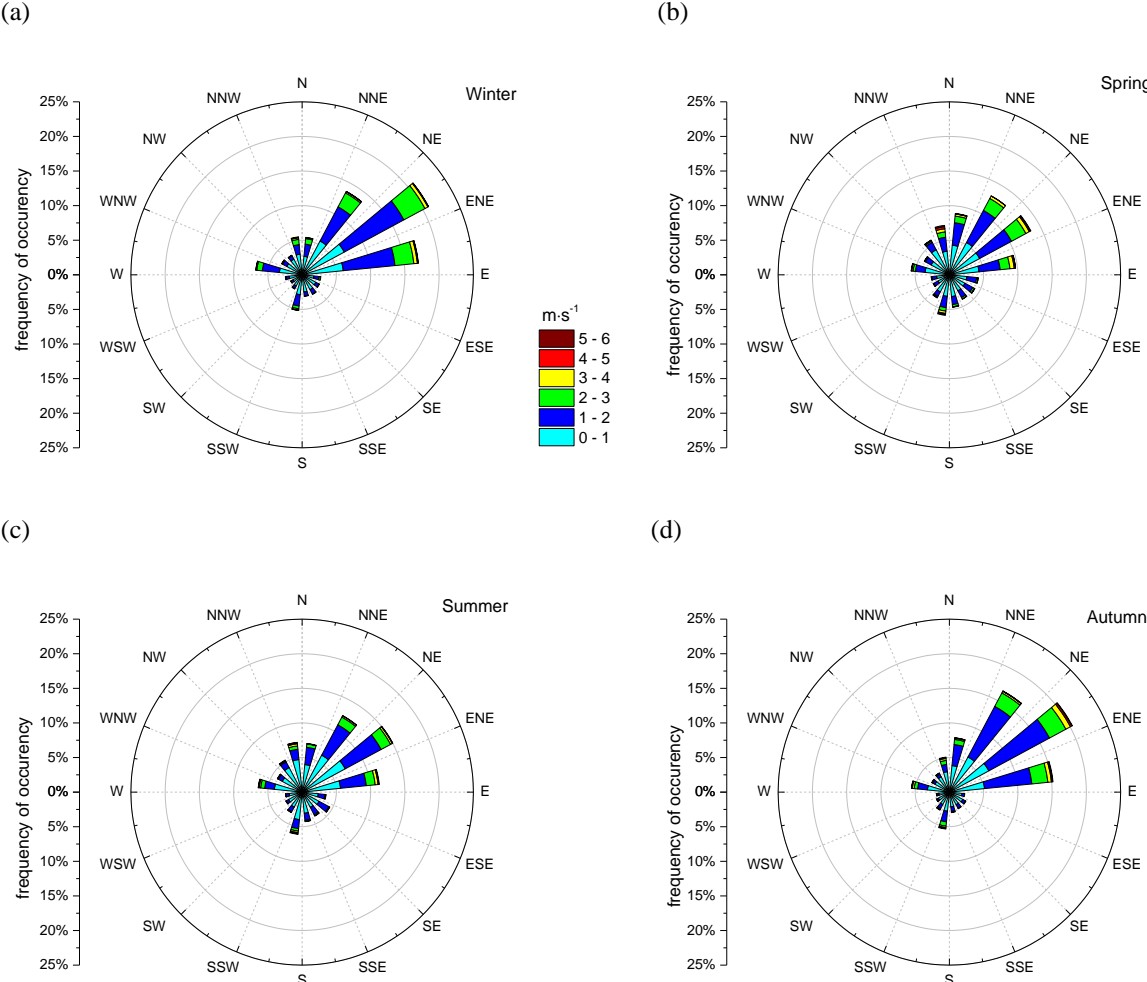

5  **Figure 22: Wind rose diagrams based on all hourly averaged observations of the weather station for winter (a), spring (b), summer (c) and autumn (d) from 2011 to 2014.**



(a)

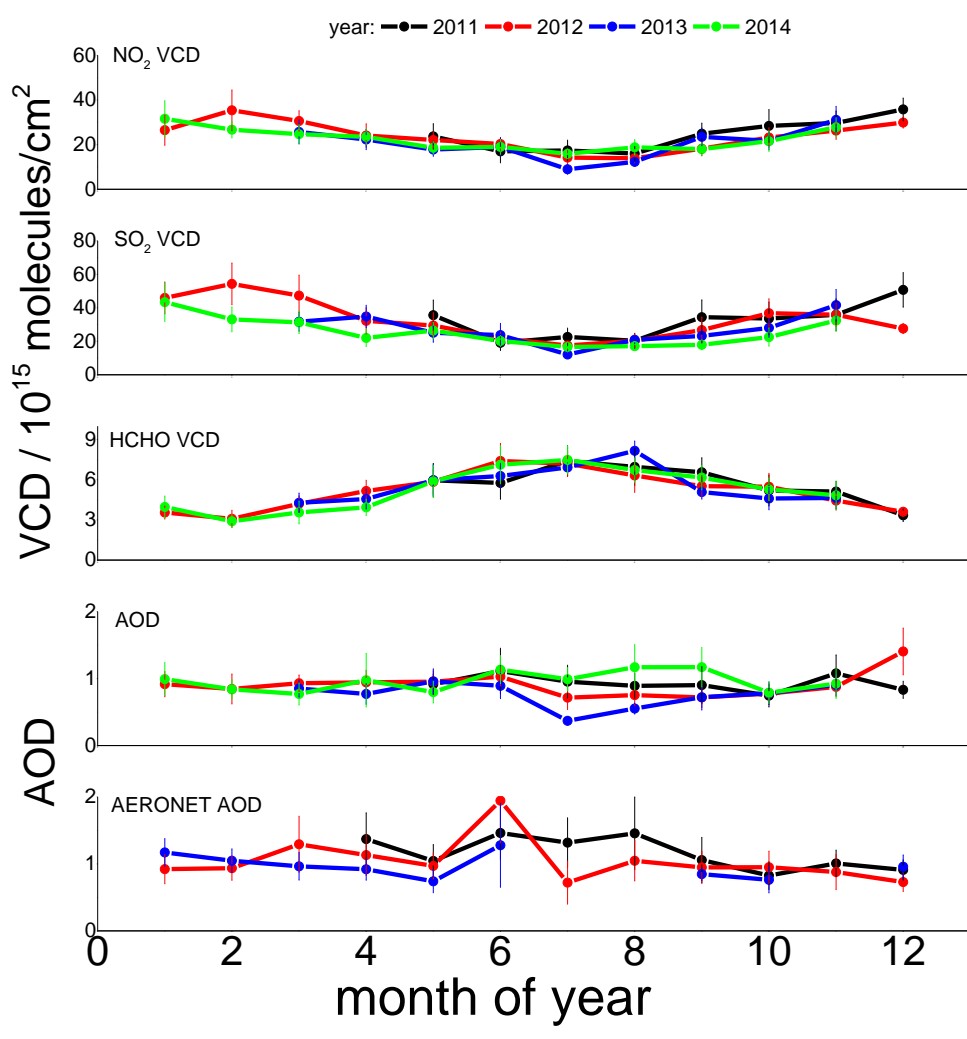





(b)

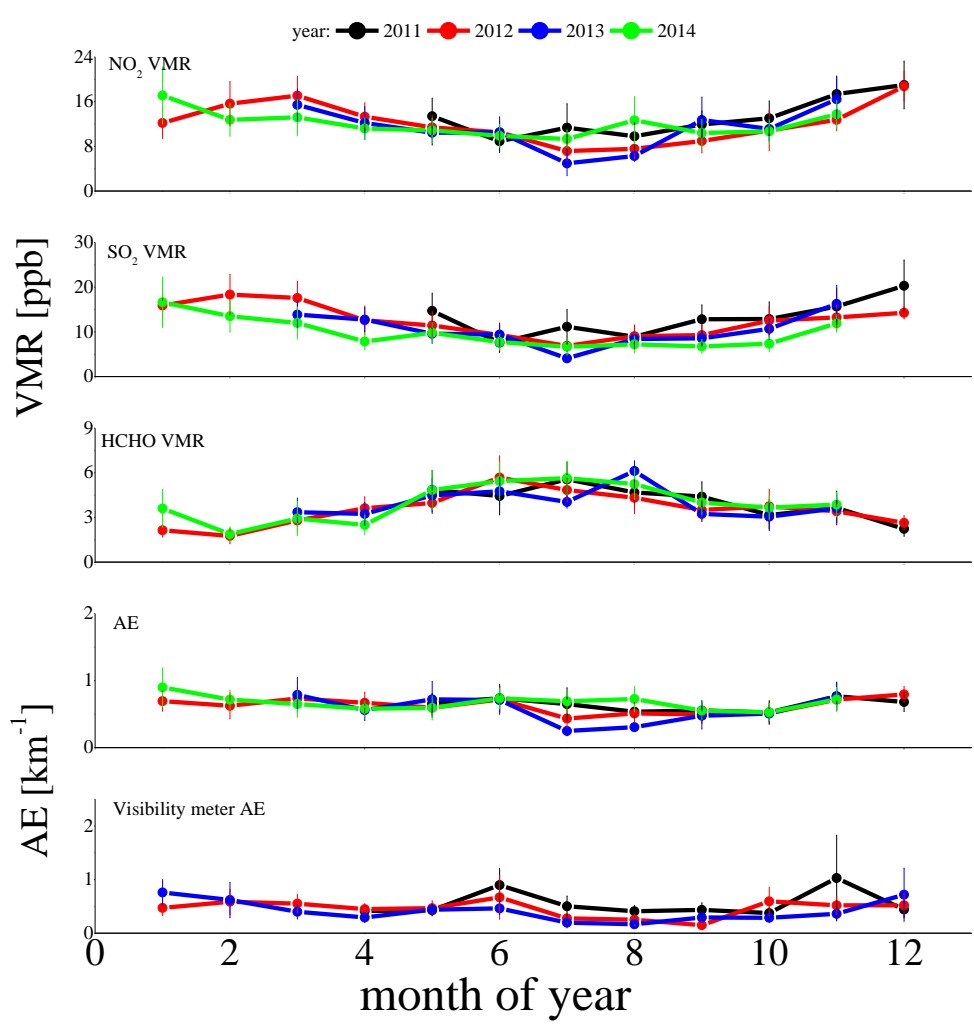





(c)



**Figure 23: Seasonal cycle of monthly mean MAX-DOAS results: VCD and AOD (a), and near-surface VMR of NO₂, SO₂ and HCHO and AE (b) for May 2011 to November 2014. The error bars represent the standard deviations. In addition to the MAX-DOAS data also AOD and AE from AERONET and visibility meter are shown, respectively. The numbers of available days in each month for MAX-DOAS measurements, AERONET and visibility meter are shown in subfigure (c). The different numbers of available AOD and trace gas data derived from MAX-DOAS are caused by the filter scheme (see Table 3).**





**Figure 24: Mean (May to November) VCDs (a) and near surface VMRs (b) of NO₂, SO₂, HCHO as well as AODs and near surface aerosol extinctions (c) for each year.**





(a)

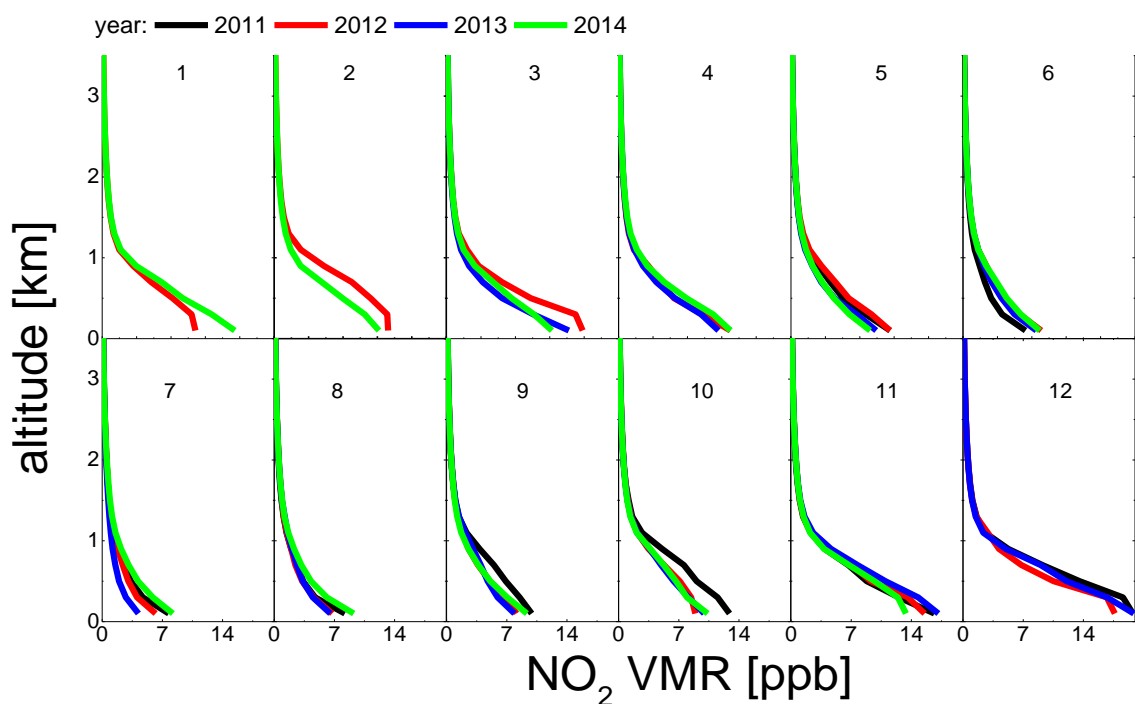

(b)

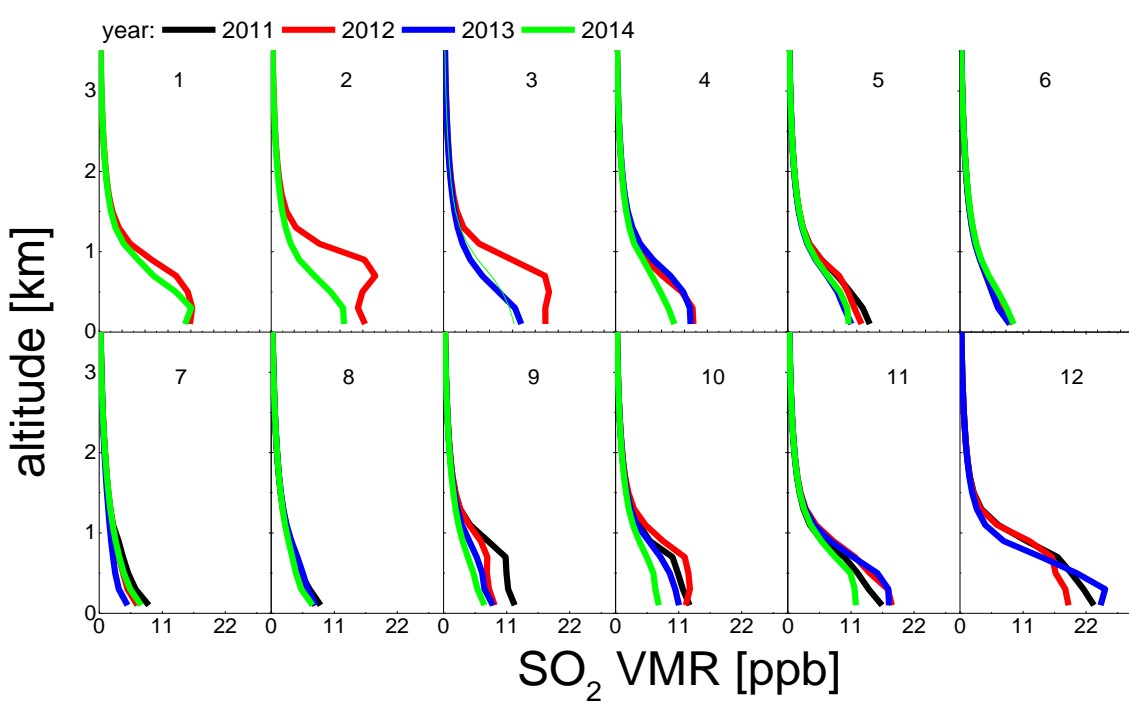





(c)

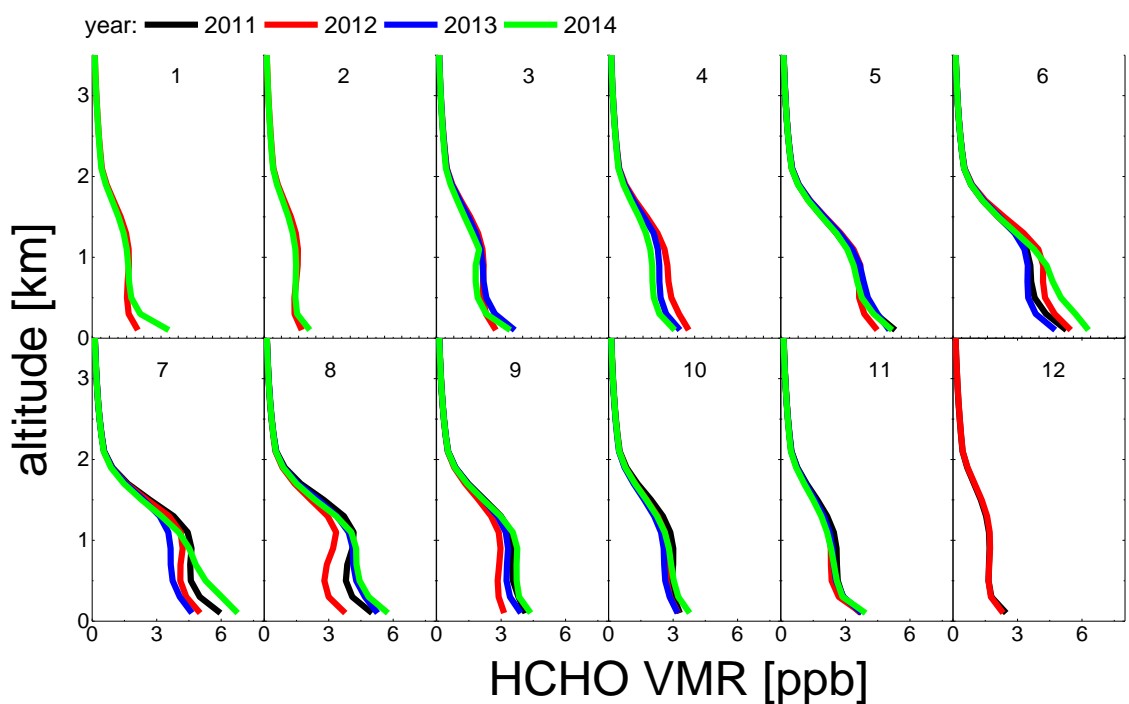

(d)

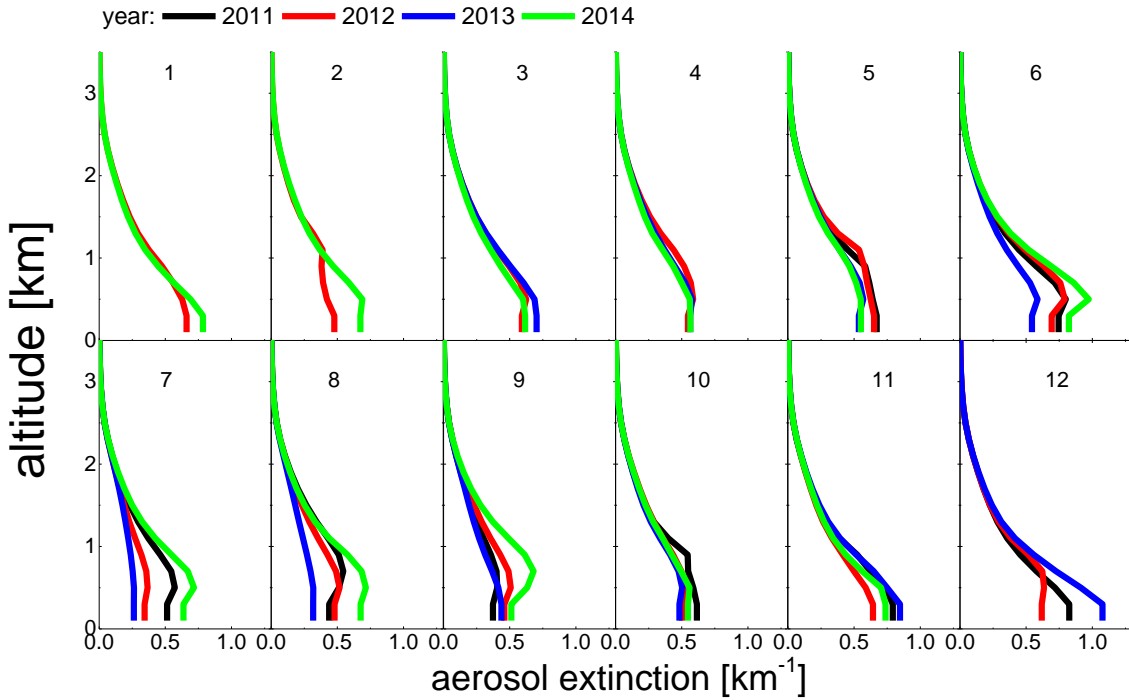





**Figure 25: Monthly mean profiles of NO₂ (a), SO₂ (b), HCHO (c) VMRs (under clear and cloudy sky conditions except thick clouds and fog) and aerosol extinction (under clear sky conditions) (d) for May 2011 to November 2014.**

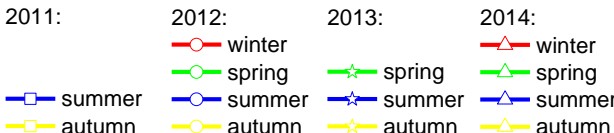

(a)

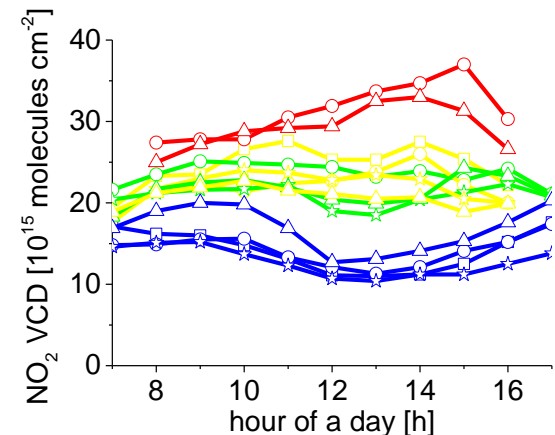

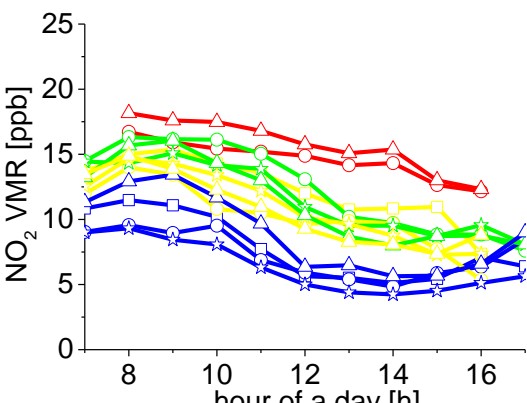

(b)

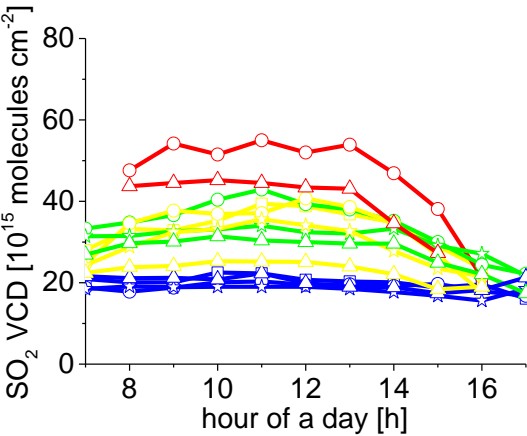

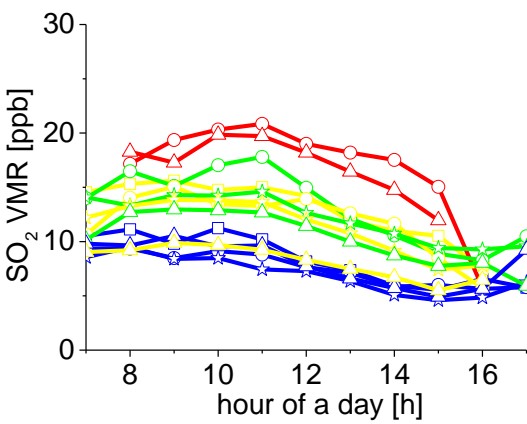





(c)

(d)

**Figure 26: Seasonally averaged diurnal variations of TG VCDs and AOD (left) and near surface values (right) of NO₂ (a), SO₂ (b), HCHO (c) and aerosols (d) from 2011 to 2014.**





**Figure 27: Mean weekly cycles of VCDs (a) and near-surface VMRs (b) of NO₂, SO₂ and HCHO as well as the AODs and near-surface AEs (c) for all MAX-DOAS observations from 2011 to 2014. The dashed lines denote the mean values during the working days from Monday to Friday (same colours as for the daily averages).**





(a)

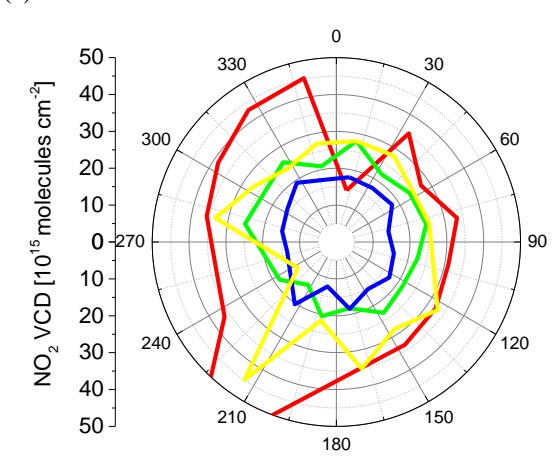
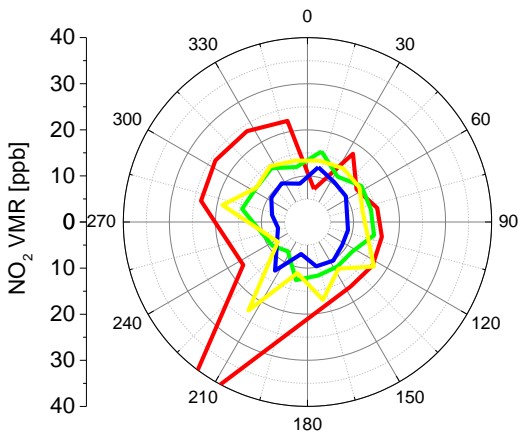

(b)

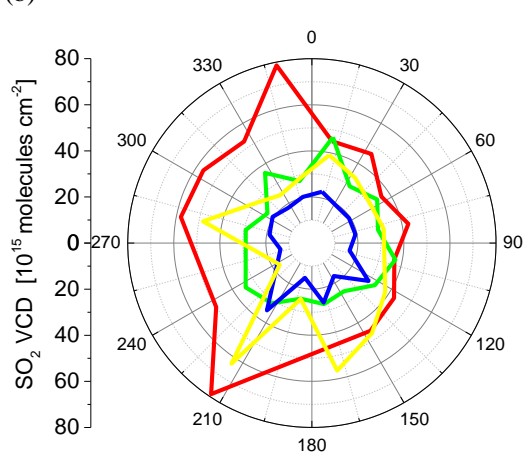
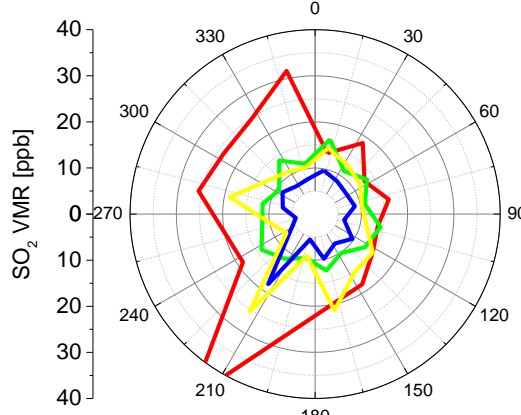



(c)

(d)

5 **Figure 28: Dependencies of VCDs and AODs (left) and near-surface VMRs and AEs (right) of NO₂ (a), SO₂ (b), HCHO (c) and aerosols (d) on wind directions for individual seasons (different colours).**



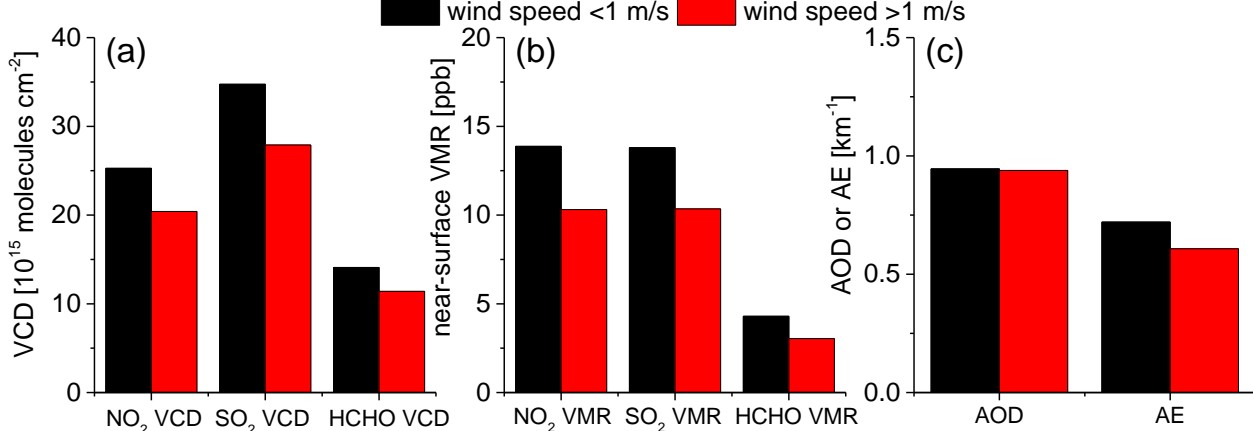

**Figure 29: Comparisons of VCDs (a) and near-surface VMRs (b) of NO₂, SO₂ and HCHO, as well as AODs and near-surface AEs for different wind speeds (smaller than 1m/s or larger than 1m/s).**