# Peer review of "Ground-based MAX-DOAS observations of tropospheric aerosols, NO2, SO2 and HCHO in Wuxi, China, from 2011 to 2014"

_Atmospheric Chemistry and Physics, 2016_

## Referee Comment (RC1) · Anonymous Referee #2 · 20 Jun 2016

This paper presents the temporal variation and vertical distribution characters of the tropospheric aerosol extinction (AE) and trace gases (TGs, including NO2, SO2 and HCHO in this study) derived from relatively long-term (2011-2014) ground-based MAX-DOAS observations in the Wuxi city, located in eastern China. The authors developed a new profile inversion logarithm (PriAM) and applied it to deal with their MAX-DOAS measurement data in this study.

For the retrieval method, the authors find that large, systematic biases of the retrieved AE profiles and thus AOD can be induced if the seasonal variations of temperature and pressure are not considered. They also show that the traditional geometry approximation could lead to larger biases than the profile integration in the retrieval of

tropospheric vertical column densities (VCDs) by analyzing the separated processes in the retrieval from their MAX-DOAS measurement data. The authors further analyze the data retrieved from MAX-DOAS measurements and characterize the seasonal, diurnal and weekly variations of NO2, SO2, HCHO and aerosols. One of their results is the finding of a significant annual decreasing of SO2 for both VCD and surface mixing ratio.

The study and its results are interesting, providing important information not only for the development in MAX-DOAS technique, but also for further investigations of urban and reginal air pollution issues in eastern China. The manuscript can still be improved before publication if some methods could be described more clearly and some results be presented more concisely. Below are my comments and suggestions in detail.

Specific comments

In Sect. 2.3, the authors compare the VCD derived by integrating the retrieved vertical profile (VCD_pro) with the VCD calculated by the so-called geometry approximation (VCD_geo). The relative difference is denoted as Diff_total (see Eq. (3)). To identify the dominating difference (which they called "error") source, they split the total difference (Diff_total) between VCD_geo and VCD_pro into two parts: one is the difference between VCD_geo and VCD_geo_m, denoted as Diff_inversion (see Eq. (4)), and another is the difference between VCD_geo_m and VCD_pro, denoted as Diff_geometry (see Eq. (5)), where VCD_geo_m is calculated by applying the geometric approximation to the modelled dSCD. I would say it is a good idea to make such a trial. But one should be noted that the VCD_pro has been taken as a standard value in the comparison and the retrieved profile is assumed to be true in their evaluation. Are the results shown in Fig. 20 for all the measurements including cloudy and haze-foggy conditions? Since both the VCD_pro and VCD_geo_m are calculated from the retrieved profile, a large bias in the retrieved profile may lead to biases in both Diff_inversion and Diff_geometry as well as their relative contributions.

Sect. 3.5.2 can be omitted as some discussions are very speculative and the meaningfulness of results seems to be local. The measurements site of this study were made at a suburban site, located in the industrial area. From Fig 1 and Fig. 23, one can see that the dominating winds come from the NE, which is neither in the direction of Wuxin urban center nor in the direction of the large industrial sources. There is no doubt that pollution plumes from the urban center and the larger industrial sources could affect the measurement site, as shown in Fig. 22. Since the characters of the emission sources from the urban area, including the fractions of pollutants and emission heights, can be rather different from those from the industrial area, the concentrations of trace gases, aerosols and its components as well as their vertical distribution might be different for different plumes. These episode effects can be investigated in-depth explicitly in the future studies considering the focus and length of the paper .

Technical issues

P1, L15-16: Change "spatial distribution" to "vertical distribution" and remove "using vertical profiles". The measurements were made only at one station and might not be used to characterize the (3-D or 2-D) spatial distribution.

P1, L28: Change "from the aerosol results" to "for the aerosol results".

P1, L30 – P2, L2: The sentences here need to be rewritten. The phrase "are found" or "is found" occurs so many times here. Better to use them only for the most important findings. The result on wind direction dependency can be skipped as it is only locally meaningful with little information for general chemistry and transport.

P2, L8-9: Add "respectively" after "nitrate and sulfate". Remove "and methane" as methane also belongs to VOCs.

P2, L18: Actually, photochemistry of precursor gases was not discussed in the paper of Huang et al. (2014).

P2, L23: Change "Since about 15 years" to "Since about 15 years ago".

P3, L10: The "stability" and "flexibility" issues can be explained a little bit, taking the OE and look-up table methods as example.

P4, L9-12: Use "Section" or "Sect.", and the same for other places in the manuscript.

P4, L13: Change "discussed" to "summarized".

P5, L13-17: I would suggest to rewrite this paragraph as "The PriAM algorithm was originally introduced by Wang et al. (2013a and b). Below we summarize the basic concept of the PriAM algorithm and its implementation settings for this study, while details can be found in Sect. 2 of the supplement. Like for other algorithms, a two-step inversion procedure is used in PriAM. In the first step, tropospheric vertical profiles (in the layer from the ground to the altitude of 4 km) of aerosol extinction are retrieved from the O4 dSCDs. Afterwards, the profiles of NO2, SO2 and HCHO volume mixing ratios (VMRs) are retrieved from the respective dSCDs in each MAX-DOAS elevation angle sequence".

P5, L24: "Fig. 7" appears earlier than "Fig. 3".

P5, L29: To do (simulate) what with RTM?

P5, L32: For the single scattering albedo, a fixed value of 0.9 is used, or it is allowed to change between 0.85 and 0.95 in the retrievals?

P6, L18: Fig. S10 should be relabeled as Fig. S8 (its position should also be moved to the front in the Supplement).

P7, L24: Change "shown" to "as shown".

P8, L16: How are clear sky conditions classified, by AERONET data or by MAX-DOAS data?

P9, L24-25: It makes me confusing that the retrieval is based on a "forward model". Do you mean a "radiation transfer" model or you still have a "backward" model?

P11, L12-13: I would suggest moving Figs. 16-19 to the Supplement.

P12, L10: Change "45 k" to "45 K".

P14, L5-13: Sect. 3.1 can be skipped.

P15, L1-2: This sentence can be rewritten, e.g., as "The observed seasonal variations of the different species are related to various processes: the seasonal variations of source emissions, chemical formation and destruction, dry and wet deposition, and atmospheric transport".

P15, L10-12: It is difficult to understand that there is no seasonal variation for NOx while the SO2 emissions vary by about 20%. Note that the boiler for domestic heating could also make a contribution to the NOx emissions.

P17, L33 - P18, L3: Similar to the seasonal variation (P15, L1-2), the diurnal variation can be affected by various factors. The explanations here seems to be very speculative and can be skipped.

P18, L16: The title of Sect. 3.5.1 can be omitted since Sect. 3.5.2 has been suggested to be skipped.

P18, L17-19: I would suggest to skip over the sentences "Huang et al. (2014) ......
The aerosol in Wuxi close to Shanghai is expected to have similar properties". There are many kinds of properties for aerosols. It is not clear what properties of aerosols are referred to here. The statement that the aerosols in Wuxi have similar properties with those in Shanghai is very speculative.

P19, L18-21: Too speculative.

P20, L2: Change "3 km" to "4 km".

P21, L18-24: This paragraph can be omitted.

P22-30: Use indented lines for each reference.

P31, Table 1: The format of this table looks not good and needs to be rearranged. Try to avoid using the same items in both column and row. For instance, use species for each column, and for the row use Cross section, Fitting interval, Polynomial degree, Intensity, and so on.

P34, L9: Change "mean maps of" to "maps of mean".

P37, Fig. 5; P39, Fig. 8; P44, Fig. 15: Better to reduce the absolute maximum/minimum values in Y-Axis appropriately so that the differences can be seen more clearly.

P39-41: Figs. 9-12 can be merged into one figure, with 4 columns for different seasons.

P41-42, Fig. 13: It might be difficult to understand the top panels (colored) of this figure if one had not read the manuscript carefully. It can be more helpful if some words like "primary sky" and "secondary sky" are added, e.g., to the legend, in the figure.

P45-46: It is suggested to move Figs. 16-19 to the Supplement.

P48-49, Figs. 21-22: These two figures can be omitted or be moved to the Supplement. Since the seasonal variability of wind directions is not so high, you may consider making a wind rose diagram averaged for all the experiment period and adding it to Fig. 1.

P58, Fig. 27: The positions of the characters in X-Axis need to be adjusted.

P59-60, Fig. 28: This figure can be omitted or be moved to the Supplement.

Supp.-P1, L31: With what do NO2 and O4 dSCDS show an systematic increase or decrease?

Supp.-P3, L12: The dSCDs shown here read not as large as two times of the mean RMS.

Supp.-P6, L16: Fig. S9 should be renumbered as its position be moved the place after

Fig. S23. Change "the for elevation angles" to " the elevation angles".

Supp.-P7, Fig. S1; P8, Fig. S2; Fig. 10, Fig. S5: Both RAA and SAA are used. Please check if they refer to the same variable.

Supp.-P12, Fig. S8: I did not find a place in the main manuscript as well as in the Supplement that this figure is referred to.

Supp.-P15, Fig. S10: This figure should be moved to the front.

Supp.-P19, L3: Change "ds" to "DoF"?

Supp.-P7-37: Please try to let the main body figure and its caption to be in the same page.

Supp.-P39: Use indented lines for each reference.

---

## Referee Comment (RC2) · Anonymous Referee #1 · 10 Aug 2016

This paper presents long-term (May 2011-November 2014) MAX-DOAS observations of tropospheric aerosols, NO2, SO2 and HCHO in Wuxi, China. Vertical profiles of trace gas concentrations and aerosol extinctions are retrieved using a new inversion algorithm called PriAM. It is based on the Levenberg-Marquardt modified Gauss-Newton numerical procedure and uses the SCIATRAN radiative transfer model as forward model. In the first part of the paper, MAX-DOAS observations and the PriAM algorithm are described and the following issues are investigated: impact of the surface pressure and temperature seasonality regularly observed in Wuxi on O4 VCD and aerosol retrieval, observed differences between VCDs derived using the geometric approximation and the profiling algorithm, impact of the sky conditions on the aerosol and trace

gases retrievals and on the agreement between MAX-DOAS and correlative measurements (AERONET, LP-DOAS). In the second part of the paper, MAX-DOAS data are used to characterize the seasonal, diurnal, and weekly variations of NO2, SO2, HCHO, and aerosols.

This is a very interesting study of high scientific quality which fits well with the scope of ACP. I recommend its final publication after addressing the following comments:

Major Comment:

Although clearly structured, the manuscript is difficult to read due to the large number of figures and panels in the manuscript itself (29) and in the supplement (28). The authors should improve the readability of their paper for the final publication in ACP by focussing on the main results only and asking themselves which figures are needed to best illustrate these results.

Specific Comments:

1) Sect. 2.2.3, page 6: One important result is the impact on the seasonal variation of the pressure and temperature profiles on the retrieved AODs and aerosol extinctions. So far most MAXDOAS groups were using US Standard Atmosphere in their aerosol extinction profile retrievals from O4 slant column densities. The authors show in their study that ignoring this systematic seasonal variation in Wuxi can cause a 20-30% bias on the retrieved AODs and near-surface aerosol extinctions, possibly yielding to unrealistic seasonal variations of these quantities. Is such a large effect only specific to the Wuxi region or can we expect similar features in other parts of China, especially the Beijing area where several MAXDOAS instruments are currently in operation ?

2) Sect. 2.2.4, page 6 and Sect. 2.2.5, page 8: The evaluation of the internal consistency of the inversion algorithm and the validation of the retrieval results are performed for favourable measurement conditions, i.e. 'clear-sky with relatively low aerosols (average AOD of about 0.6)'. 'average AOD of about 0.6' is for me very vague. Is it the

daily or hourly average ? Looking at Figure 9 showing the scatterplots of the MAX-DOAS versus AERONET AODs, MAXDOAS AOD values much larger than 0.6 (see scatterplot for Spring) are selected while this figure is supposed to illustrate the agreement between MAXDOAS and AERONET in low aerosols (and clear-sky) conditions. I think a clarification is needed here.

3) Sect. 2.2.5, page 8: MAXDOAS AODs are validated using AERONET data from the Taihu station, which is located at 18 km south west of the MAXDOAS instrument location. Based on the coordinates mentioned on the AERONET website, the sun photometer is located in the south of the Taihu mountains, at the edge the Taihu Lake (west of the lower left corner of the orange rectangle in Figure 1b), which seems to be a much more remote area than the one from where the MAXDOAS instrument is operating (see also figure 1b). The question is therefore how representative is the Taihu station compared to the location of the MAXDOAS instrument ? Did the authors consider this point for the interpretation of their comparison results ?

4) Sect. 2.2.5, pages 8-9: the validation results are discussed only in terms of absolute differences between MAXDOAS and correlative data. It would be useful for the reader to have also an idea about the corresponding relative difference values.

5a) Sect. 2.2.6, pages 9-10: Is it really useful to show the scatterplots of MAXDOAS versus correlative data and histograms of the absolute differences between MAXDOAS and correlative data for all sky conditions, seasons, and trace gas (TG) and aerosol variables (Figs 15-19 and S20-23) ? I think the manuscript could be simplified here. My suggestion for the final publication is to remove all the histograms from the manuscript and to present the linear regression and correlation results in a table. The latter could also be presented in a panel like the ones usually used in MAXDOAS intercomparison campaigns for summarizing the slant column density comparison results (see e.g. Figure 6 in Roscoe et al., Atmos. Meas. Tech., 3, 1629–1646, 2010; the authors would show for each TG or aerosol variable, the correlation coefficient, slope, and intercept values in three different subplots, replacing the x-axis by the sky conditions and the

elevation angles (colored circles) by the seasons).

5b) Based on these comparison results (Figs 15-19 and S20-23), the authors have developed recommendations for determining under which sky conditions which TG and aerosol data products can be used or not. This filter scheme is presented in Table 3. However, nothing is said on the criteria (e.g. which threshold values on correlation coefficients and/or slopes, etc) used in practice by the authors to develop these recommendations. Maybe a panel summarizing the linear regression results as suggested in comment 5a could also support the discussion here.

6) Sect. 2.2.7, page 12, lines 12-13: the total error budgets of aerosol retrievals are simply reported as error of TGs related to the errors of aerosols. This is not correct because the relationship between the aerosol extinction profiles and the TG retrievals is not linear, i.e. a 15% difference in the aerosol extinction profile used in the TG retrievals does not lead necessarily to a 15% difference in the retrieved TG profiles. This point should be further investigated and corrected in the revised manuscript.

7) Section 3.2: Interpreting the retrieved profile shapes should be done with caution given the fact that the average DFS is only around 2. This particularly the case for the HCHO profiles which show a secondary maximum around 1km. A possible explanation is of course the transport of longer-lived VOCs to higher altitudes but with a DFS around 2, one should not totally exclude the possibility of a retrieval artifact.

Suggestions for technical corrections:

*Page 1, line 16: 'extinctions' -> 'extinction'

*Page 2, line 26: 'ground based' -> 'ground-based'; this should be corrected throughout the manuscript.

*Page 3, line 8: 'Clemer' -> 'Clémer'; this should be corrected throughout the manuscript

*Page 3, line 22: 'humidity controlled' -> 'humidity-controlled'

*Page 3, line 34: 'so called' -> 'so-called'

*Page 5, line 26: 'long term' -> 'long-term'

*Page 6, line 11: 'session' -> 'section'

*Page 12, line 21: 'budges' -> 'budget' or 'budgets'

*Page 18, line 10: remove '.' after 'found:'

*Page 38, Figure 7: the '2' of 'NO2' on the x-axis label of subplot (b) is cut.

*Page 39, Figure 9: remove the ',' after 'Figure 9:'

*Supplement, page 8, Figure S2: There is a problem with y-axis labels of subplots (a) and (b).

---

## Author Comment (AC1) · 18 Oct 2016

Reply to Ref. #1

First of all we want to thank this reviewer for the positive assessment of our manuscript and the constructive and helpful suggestions.

General comments This paper presents long-term (May 2011-November 2014) MAX-DOAS observations of tropospheric aerosols, NO2, SO2 and HCHO in Wuxi, China. Vertical profiles of trace gas concentrations and aerosol extinctions are retrieved using a new inversion algorithm called PriAM. It is based on the Levenberg-Marquardt modified Gauss-Newton numerical procedure and uses the SCIATRAN radiative transfer model as forward model. In the first part of the paper, MAX-DOAS observations and the PriAM algorithm are described and the following issues are investigated: impact of the surface pressure and temperature seasonality regularly observed in Wuxi on O4 VCD and aerosol retrieval, observed differences between VCDs derived using the geometric approximation and the profiling algorithm, impact of the sky conditions on the aerosol and trace gases retrievals and on the agreement between MAX-DOAS and correlative measurements (AERONET, LP-DOAS). In the second part of the paper, MAX-DOAS data are used to characterize the seasonal, diurnal, and weekly variations of NO2, SO2, HCHO and aerosols. This is a very interesting study of high scientific quality which fits well with the scope of ACP. I recommend its final publication after addressing the following comments.

Author reply: Many thanks for the positive assessment!

Major comments

-Although clearly structured, the manuscript is difficult to read due to the large number of figures and panels in the manuscript itself (29) and in the supplement (28). The authors should improve the readability of their paper for the final publication in ACP by focusing on the main results only and asking themselves which figures are needed to best illustrate these results.

Author reply: We minimized the number of figures to only show the necessary figures in the main manuscript. In the revised version there are only 15 figures in the main part of the manuscript and in total fewer subplots in the manuscript and supplement than in the original version. To make the paper more focused on the most important new findings, we moved the original section 2.2.4 into the supplement as new section 3.2 (the main conclusion of the original section 2.2.4 is summarized at the end of section 2.2.2. We also moved the original Fig. 5 into the supplement as new Fig. S10. The original Fig. S11 is removed from the supplement. We made the new Fig. 6 in order to replace the original Figs. 8-12. The key information including the mean differences, standard

deviations, R, slopes, intercepts and numbers of observations in the original figures are now all plotted in the new Fig. 6. We made the new Fig. 9 in order to replace the original Figs. 15-19 to only show the key information. In the new supplement the new Figs. S21-24 summarize the key information from the previous Figs. S20-S23. We also moved the original section 3.1 and Fig. 21 and 22 (containing the meteorology data) into the supplement (a yellow arrow is added in Fig.1b of the revised version to show the dominant wind direction). We also removed the original Fig. 4a and only keep Fig. 4b in the revised version (because the information in Fig. 4a is already well presented in Fig. 4b). The original Fig. 7 is removed. Although the current supplement still contains many figures, we think it is good for the readers who want to learn the details of the study. Thus we paid more effort to shorten the main manuscript.

Specific Comments:

1) Sect. 2.2.3, page 6: One important result is the impact on the seasonal variation of the pressure and temperature profiles on the retrieved AODs and aerosol extinctions. So far most MAXDOAS groups were using US Standard Atmosphere in their aerosol extinction profile retrievals from O4 slant column densities. The authors show in their study that ignoring this systematic seasonal variation in Wuxi can cause a 20-30% bias on the retrieved AODs and near-surface aerosol extinctions, possibly yielding to unrealistic seasonal variations of these quantities. Is such a large effect only specific to the Wuxi region or can we expect similar features in other parts of China, especially the Beijing area where several MAXDOAS instruments are currently in operation?

Author reply: The variation of O4 VCD depends on the systematic seasonal variation of temperature and pressure. The seasonal variation of temperature occurs in many locations of the world (especially outside the tropics). However the temporal variation of the pressure is usually more complex and can be very different at different locations. The variation of pressure in Wuxi (and also many other parts of Eastern China) is related to the East Asian Monsoon and shows a systematic seasonal pattern. The monsoon is a general phenomenon in the eastern China. The pressure in the conti-

nent is systematically lower and higher than that in the ocean in summer and winter, respectively. Thus a similar seasonal variation of the O4 VCD is expected in general in Eastern China including Beijing. We added this information at the end of section 2.2.3 of the revised manuscript.

2) Sect. 2.2.4, page 6 and Sect. 2.2.5, page 8: The evaluation of the internal consistency of the inversion algorithm and the validation of the retrieval results are performed for favourable measurement conditions, i.e. 'clear-sky with relatively low aerosols (average AOD of about 0.6)'. 'Average AOD of about 0.6' is for me very vague. Is it the daily or hourly average? Looking at Figure 9 showing the scatterplots of the MAX-DOAS versus AERONET AODs, MAXDOAS AOD values much larger than 0.6 (see scatterplot for Spring) are selected while this figure is supposed to illustrate the agreement between MAXDOAS and AERONET in low aerosols (and clear-sky) conditions. I think a clarification is needed here.

Author reply: Thanks for pointing out the misleading description. 'Clear-sky with relatively low aerosols' belongs to one category of the sky conditions identified by MAX-DOAS observations. Please see section 2.2.5. To clarify the point, we added a sentence at the end of the first paragraph of section 2.2.5: "Another point which needs to be clarified is that distinguishing "low aerosols" and "high aerosols" is based on the colour index observed by MAX-DOAS. Thus there is not an explicit AOD value which distinguishes both aerosol categories. The studies of Wang et al., 2015, however, demonstrated that the AODs observed by the Taihu AERONET sun photometer are mostly smaller and larger than 0.6 for the "low aerosols" and "high aerosols", respectively. In addition to the cloud effect, also the effect of high aerosol loads is evaluated (due to the unrealistic assumption of the pdf of the atmospheric state in the OE algorithm for high aerosol loads (see Eq. (1))." Note that in the revised version of the manuscript the original section 2.2.4 was moved to the supplement as new section 3.2. In the beginning of section 3.2 in the supplement, we deleted "average AOD of about 0.6", but write "the sky condition is directly identified by MAX-DDOAS observations,

see section 2.2.5".

3) Sect. 2.2.5, page 8: MAXDOAS AODs are validated using AERONET data from the Taihu station, which is located at 18 km south west of the MAXDOAS instrument location. Based on the coordinates mentioned on the AERONET website, the sun photometer is located in the south of the Taihu mountains, at the edge the Taihu Lake (west of the lower left corner of the orange rectangle in Figure 1b), which seems to be a much more remote area than the one from where the MAXDOAS instrument is operating (see also figure 1b). The question is therefore how representative is the Taihu station compared to the location of the MAXDOAS instrument? Did the authors consider this point for the interpretation of their comparison results?

Author reply: Thanks for pointing out this potential source for differences. We added the following discussion to section 2.2.4: "Here it should be noted that AERONET Taihu station is located in a more remote area (from downtown Wuxi) than the MAX-DOAS at Wuxi station. The different locations could contribute to a systematic bias between both data sets. However the long residence time of up to several days (Ahmed et al., 2004) and the relatively homogeneous horizontal distribution of aerosols (implied by the weak dependence of AOD on wind direction, see section 3.4.2) implies that the differences between both measurements should be small."

4) Sect. 2.2.5, pages 8-9: the validation results are discussed only in terms of absolute differences between MAXDOAS and correlative data. It would be useful for the reader to have also an idea about the corresponding relative difference values.

Author reply: We added the relative differences (compared to the average values) to the manuscript in section 2.2.4 and the conclusions.

5) a) Sect. 2.2.6, pages 9-10: Is it really useful to show the scatterplots of MAXDOAS versus correlative data and histograms of the absolute differences between MAXDOAS and correlative data for all sky conditions, seasons, and trace gas (TG) and aerosol variables (Figs 15-19 and S20-23) ? I think the manuscript could be simplified here. My

suggestion for the final publication is to remove all the histograms from the manuscript and to present the linear regression and correlation results in a table. The latter could also be presented in a panel like the ones usually used in MAXDOAS intercomparison campaigns for summarizing the slant column density comparison results (see e.g. Figure 6 in Roscoe et al., Atmos. Meas. Tech., 3, 1629–1646, 2010; the authors would show for each TG or aerosol variable, the correlation coefficient, slope, and intercept values in three different subplots, replacing the x-axis by the sky conditions and the elevation angles (colored circles) by the seasons).

Author reply: Great thanks for your good suggestions! We followed your suggestions to re-plot the figures. Please see the description in the reply to your "Major comments".

b) Based on these comparison results (Figs 15-19 and S20-23), the authors have developed recommendations for determining under which sky conditions which TG and aerosol data products can be used or not. This filter scheme is presented in Table 3. However, nothing is said on the criteria (e.g. which threshold values on correlation coefficients and/or slopes, etc) used in practice by the authors to develop these recommendations. Maybe a panel summarizing the linear regression results as suggested in comment 5a could also support the discussion here.

Author reply: It is hard to quantify the cloud effects on the MAX-DOAS results of aerosols and TGs. Here we combine the results shown in Fig. 8 and 9 (in the revised version) to qualitatively discuss the effect of clouds and give our recommendation. We added more discussion of the cloud effects (as shown in the three figures in the end two paragraphs of section 2.2.5) in order to make our selection more clear.

6) Sect. 2.2.7, page 12, lines 12-13: the total error budgets of aerosol retrievals are simply reported as error of TGs related to the errors of aerosols. This is not correct because the relationship between the aerosol extinction profiles and the TG retrievals is not linear, i.e. a 15% difference in the aerosol extinction profile used in the TG retrievals does not lead necessarily to a 15% difference in the retrieved TG profiles.

This point should be further investigated and corrected in the revised manuscript.

Author reply: We agree with the reviewer. But the quantification of the aerosol effects on TG results is difficult because it depends on the aerosol profile, aerosol properties, profiles of TGs and even observation geometries. Many simulation studies need to be done to acquire a more reasonable estimation on the effects. Although the topic is very interesting and new, but it should be done in a separated work in the future. Thus we would only coarsely estimate the relevant errors of TGs using the assumption of linear propagation of the errors of aerosol retrievals. And the following clarification is given in the manuscript: "The estimations of aerosol relevant errors are rough. A further studies need to be done to acquire a more reasonable estimation by considering aerosol properties, profiles of aerosols and TGs and observation geometries."

7) Section 3.2: Interpreting the retrieved profile shapes should be done with caution given the fact that the average DFS is only around 2. This particularly the case for the HCHO profiles which show a secondary maximum around 1km. A possible explanation is of course the transport of longer-lived VOCs to higher altitudes but with a DFS around 2, one should not totally exclude the possibility of a retrieval artifact.

Author reply: We agree with the reviewer. We added the following clarification in section 3.1 of the revised version: "However it should be noted that VMRs of HCHO at high altitudes are strongly constrained by the a-priori profiles because of the low sensitivity of MAX-DOAS retrievals at these altitudes. More comparisons studies with aircraft measurements need to be done in the future to further quantify the retrieval sensitivities for elevated layers." In my own opinion, we still have confidence on the extensive vertical distribution of HCHO retrieved by MAX-DOAS because of two reasons: 1) the Fig. S9 in the supplement of the revised version indicates the higher vertical extension can still be partly represented even for using an exponential a-priori profile; 2) the large variation amplitude of HCHO VMRs at the altitude around 1km is retrieved from MAX-DOAS observations. It indicates the sensitivity of MAX-DOAS retrievals to the layers at around 1km is still well. Thus we add the following comment in section 3.1:

[Figure]

"Nevertheless we still have confidence on the extensively vertical distribution of HCHO retrieved by MAX-DOAS because of two reasons: 1) the Fig. S9 in the supplement indicates the higher vertical extension can be partly represented even for using an exponential a-priori profile; 2) the large variability of HCHO VMRs at the altitude around 1km is retrieved from MAX-DOAS observations. It indicates the sensitivity of MAX-DOAS retrievals to the elevated layers is still well."

Suggestions for technical corrections:

*Page 1, line 16: 'extinctions' -> 'extinction'

Corrected

*Page 2, line 26: 'ground based' -> 'ground-based'; this should be corrected throughout the manuscript.

Corrected

*Page 3, line 8: 'Clemer' -> 'Clémer'; this should be corrected throughout the manuscript

Corrected

*Page 3, line 22: 'humidity controlled' -> 'humidity-controlled'

Corrected

*Page 3, line 34: 'so called' -> 'so-called'

Corrected

*Page 5, line 26: 'long term' -> 'long-term'

Corrected

*Page 6, line 11: 'session' -> 'section'

Corrected

*Page 12, line 21: 'budges' -> 'budget' or 'budgets'

Corrected

*Page 18, line 10: remove '.' after 'found:'

Corrected

*Page 38, Figure 7: the '2' of 'NO2' on the x-axis label of subplot (b) is cut.

Corrected

*Page 39, Figure 9: remove the ',' after 'Figure 9:'

Corrected

*Supplement, page 8, Figure S2: There is a problem with y-axis labels of subplots (a) and (b).

Corrected

---

## Author Comment (AC2) · 18 Oct 2016

Reply to Ref. #2

First of all we want to thank this reviewer for the positive assessment of our manuscript and the constructive and helpful suggestions.

General comments This paper presents the temporal variation and vertical distribution characters of the tropospheric aerosol extinction (AE) and trace gases (TGs, including NO2, SO2 and HCHO in this study) derived from relatively long-term (2011-2014) ground-based MAX-DOAS observations in the Wuxi city, located in eastern China. The authors developed a new profile inversion logarithm (PriAM) and applied it to deal with

their MAX-DOAS measurement data in this study. For the retrieval method, the authors find that large, systematic biases of the retrieved AE profiles and thus AOD can be induced if the seasonal variations of temperature and pressure are not considered. They also show that the traditional geometry approximation could lead to larger biases than the profile integration in the retrieval of tropospheric vertical column densities (VCDs) by analyzing the separated processes in the retrieval from their MAX-DOAS measurement data. The authors further analyze the data retrieved from MAX-DOAS measurements and characterize the seasonal, diurnal and weekly variations of NO2, SO2, HCHO and aerosols. One of their results is the finding of a significant annual decreasing of SO2 for both VCD and surface mixing ratio. The study and its results are interesting, providing important information not only for the development in MAX-DOAS technique, but also for further investigations of urban and regional air pollution issues in eastern China. The manuscript can still be improved before publication if some methods could be described more clearly and some results be presented more concisely. Below are my comments and suggestions in detail.

Author reply: Many thanks for the positive assessment! In order to present the results more concisely, we minimized the number of figures to only show the necessary figures in the main manuscript. In the revised version there are only 15 figures in the main part of the manuscript and in total fewer subplots in the manuscript and supplement than in the original version. To make the paper more focused on the most important new findings, we moved the original section 2.2.4 into the supplement as new section 3.2 (the main conclusion of the original section 2.2.4 is summarized at the end of section 2.2.2. We also moved the original Fig. 5 into the supplement as new Fig. S10. The original Fig. S11 is removed from the supplement. We made the new Fig. 6 in order to replace the original Figs. 8-12. The key information including the mean differences, standard deviations, R, slopes, intercepts and numbers of observations in the original figures are now all plotted in the new Fig. 6. We made the new Fig. 9 in order to replace the original Figs. 15-19 to only show the key information. In the new supplement the new Figs. S21-24 summarize the key information from the previous Figs. S20-S23.

[Figure]

We also moved the original section 3.1 and Fig. 21 and 22 (containing the meteorology data) into the supplement (a yellow arrow is added in Fig.1b of the revised version to show the dominant wind direction). We also removed the original Fig. 4a and only keep Fig. 4b in the revised version (because the information in Fig. 4a is already well presented in Fig. 4b). The original Fig. 7 is removed. Although the current supplement still contains many figures, we think it is good for the readers who want to learn the details of the study. Thus we paid more effort to shorten the main manuscript.

Specific comments

1) In Sect. 2.3, the authors compare the VCD derived by integrating the retrieved vertical profile (VCD_pro) with the VCD calculated by the so-called geometry approximation (VCD_geo). The relative difference is denoted as Diff_total (see Eq. (3)). To identify the dominating difference (which they called "error") source, they split the total difference (Diff_total) between VCD_geo and VCD_pro into two parts: one is the difference between VCD_geo and VCD_geo_m, denoted as Diff_inversion (see Eq. (4)), and another is the difference between VCD_geo_m and VCD_pro, denoted as Diff_geometry (see Eq. (5)), where VCD_geo_m is calculated by applying the geometric approximation to the modelled dSCD. I would say it is a good idea to make such a trial. But one should be noted that the VCD_pro has been taken as a standard value in the comparison and the retrieved profile is assumed to be true in their evaluation. Are the results shown in Fig. 20 for all the measurements including cloudy and haze-foggy conditions? Since both the VCD_pro and VCD_geo_m are calculated from the retrieved profile, a large bias in the retrieved profile may lead to biases in both Diff_inversion and Diff_geometry as well as their relative contributions.

Author reply: Thanks for your general support of our approach. Concerning your suggestion, we think the problem pointed out is not that serious: if the retrieved profile is quite different from the true profile, the Diff_inversion will increase, but the Diff_geometry will not be impacted (both VCD_pro and VCD_geo_m are based on the retrieved profile). Thus the increased Diff_inversion can present the large errors of the

retrieved profile. We added this information to the end of section 2.3 of the manuscript as follows: "One point need to be clarified that the discrepancy of retrieved profile from the true profile doesn't impact the approach, although both and are as function of the retrieved profile. Because in this case, only will increase, but will not be impacted. The increased present the large errors of the profile inversion."

2) Sect. 3.5.2 can be omitted as some discussions are very speculative and the meaningfulness of results seems to be local. The measurements site of this study were made at a suburban site, located in the industrial area. From Fig 1 and Fig. 23, one can see that the dominating winds come from the NE, which is neither in the direction of Wuxi urban center nor in the direction of the large industrial sources. There is no doubt that pollution plumes from the urban center and the larger industrial sources could affect the measurement site, as shown in Fig. 22. Since the characters of the emission sources from the urban area, including the fractions of pollutants and emission heights, can be rather different from those from the industrial area, the concentrations of trace gases, aerosols and its components as well as their vertical distribution might be different for different plumes. These episode effects can be investigated in-depth explicitly in the future studies considering the focus and length of the paper.

Author reply: Although this study is local and rough, it still shows several general and important results: 1) the dependence of the measured TG VMRs on the wind direction indicates that the dominating sources of the pollutions are local, but not from the long range transport. Also, strong horizontal gradient appears. Because of the expected similar life time, meteorological conditions and emission sources, the conclusion probably fits to the whole YRD region. 2) The study provides an example on how to use ground-based MAX-DOAS observations to find strong emission sources in an urban-size area. 3) The seasonality of the wind dependence of the trace gases, especially for HCHO, indicates the different sources in different seasons. Because of these reasons, we prefer to keep this section. Further explicit studies could be done in the future to further investigate these findings. We added this statement to the end of section 3.4.2

of the revised manuscript.

Technical issues:

P1, L15-16: Change "spatial distribution" to "vertical distribution" and remove "using vertical profiles". The measurements were made only at one station and might not be used to characterize the (3-D or 2-D) spatial distribution.

Corrected

P1, L28: Change "from the aerosol results" to "for the aerosol results".

Corrected

P1, L30 – P2, L2: The sentences here need to be rewritten. The phrase "are found" or "is found" occurs so many times here. Better to use them only for the most important findings. The result on wind direction dependency can be skipped as it is only locally meaningful with little information for general chemistry and transport.

We rewrote the paragraph based on the suggestion. We prefer to keep the results about the wind direction dependency (see our reply to the second specific comment).

P2, L8-9: Add "respectively" after "nitrate and sulfate". Remove "and methane" as methane also belongs to VOCs.

Corrected

P2, L18: Actually, photochemistry of precursor gases was not discussed in the paper of Huang et al. (2014).

We corrected the sentence to "Recent studies found that in megacities in different regions of China most of aerosol particles are from secondary sources, e.g. formed through photochemistry of precursor gases, during haze pollution events (Crippa et al., 2014 and Huang et al., 2014)." Huang et al, 2014 characterized the percentages of different compositions in total aerosols.
P2, L23: Change "Since about 15 years" to "Since about 15 years ago".

Corrected

P3, L10: The "stability" and "flexibility" issues can be explained a little bit, taking the OE and look-up table methods as example

We add the explanation as "Here good stability means that an inversion approach is robust with respect to the effects of measurement noise. Good flexibility means that it can well retrieve diverse profile shapes."

P4, L9-12: Use "Section" or "Sect.", and the same for other places in the manuscript.

Corrected

P4, L13: Change "discussed" to "summarized".

Corrected

P5, L13-17: I would suggest to rewrite this paragraph as "The PriAM algorithm was originally introduced by Wang et al. (2013a and b). Below we summarize the basic concept of the PriAM algorithm and its implementation settings for this study, while details can be found in Sect. 2 of the supplement. Like for other algorithms, a two-step inversion procedure is used in PriAM. In the first step, tropospheric vertical profiles (in the layer from the ground to the altitude of 4 km) of aerosol extinction are retrieved from the O4 dSCDs. Afterwards, the profiles of NO2, SO2 and HCHO volume mixing ratios (VMRs) are retrieved from the respective dSCDs in each MAX-DOAS elevation angle sequence".

We rewrote the part based on the suggestion.

P5, L24: "Fig. 7" appears earlier than "Fig. 3".

We agree with your suggestion to correct the order. However, as described above, we already removed the Fig. 7 in the revised manuscript.

P5, L29: To do (simulate) what with RTM?

We correct it as "the RTM simulations of weighting functions are done at 370 nm".

P5, L32: For the single scattering albedo, a fixed value of 0.9 is used, or it is allowed to change between 0.85 and 0.95 in the retrievals?

A fixed value is used. To be clear, we corrected the sentence as "The fixed single scattering albedo of 0.9 and asymmetry factor (Henyey and Greenstein, 1941) of 0.72 are chosen according to average inversion results from the Taihu AERONET station from 2011 to 2013 (the data in 2014 is unavailable)."

P6, L18: Fig. S10 should be relabeled as Fig. S8 (its position should also be moved to the front in the Supplement).

We followed this suggestion. We put the Fig. S16 before Fig. S17 in the revised version.

P7, L24: Change "shown" to "as shown".

Corrected

P8, L16: How are clear sky conditions classified, by AERONET data or by MAX-DOAS data?

The sky condition is identified by MAX-DDOAS observations as described in section 2.2.5 in the revised version.

P9, L24-25: It makes me confusing that the retrieval is based on a "forward model". Do you mean a "radiation transfer" model or you still have a "backward" model?

We changed "forward model" to "RTM". Because the SCIATRAN RTM used in this study is a forward model. Both of "RTM" and forward model represent the same thing in this study.

P11, L12-13: I would suggest moving Figs. 16-19 to the Supplement.

Please see the reply to your general comment. We use the new Fig.9 in the revised version to replace the original Fig. 16-19.

P12, L10: Change "45 k" to "45 K".

Corrected

P14, L5-13: Sect. 3.1 can be skipped.

We moved section 3.1 and Fig. 21 and 22 in the revised version into the supplement. We added a yellow arrow to show the dominant wind direction in Fig. 1b.

P15, L1-2: This sentence can be rewritten, e.g., as "The observed seasonal variations of the different species are related to various processes: the seasonal variations of source emissions, chemical formation and destruction, dry and wet deposition, and atmospheric transport".

We correct it based on the suggestion as "The observed seasonal variations of the different species are related to various processes: the seasonal variation of source emissions, chemical (trans-) formation and destruction, dry and wet deposition, and atmospheric transport (Wang et al., 2010; Lin et al., 2011)".

P15, L10-12: It is difficult to understand that there is no seasonal variation for NOx while the SO2 emissions vary by about 20%. Note that the boiler for domestic heating could also make a contribution to the NOx emissions.

Because the contribution of boilers to NOx is only about 5%, but to SO2 is about 20% based on the study of Huang et al., 2011. We agree that we can't totally ignore the effect of seasonal use of domestic heating on the seasonality of NOx. But its effect on NOx is much weaker than SO2. Thus we modified the test as "It is assumed that about 94% of total NOx emission in the Wuxi region is emitted from the power plants, industrial fuel combustions and vehicles (Huang et al., 2011), which emit similar amounts in different seasons. The contribution of boilers for the seasonal use of domestic heating to NOx is only about 5% (Huang et al., 2011). Thus the seasonal variation of the MAX-

DOAS results cannot be explained by the variation of the NOx emissions. However, the SO2 emissions might vary by about 20% due to the significant contribution of boilers (Huang et al., 2011)."

P17, L33 - P18, L3: Similar to the seasonal variation (P15, L1-2), the diurnal variation can be affected by various factors. The explanation here seems to be very speculative and can be skipped.

We agree on your opinion. Thus we skip the speculative explanation and rewrite it as "Their diurnal variations can be affected by various factors, e.g. the diurnal variation of the emission sources, as well as secondary formation, deposition and dispersion."

P18, L16: The title of Sect. 3.5.1 can be omitted since Sect. 3.5.2 has been suggested to be skipped.

Because we decide to keep the section 3.4.2 in the revised manuscript (section 3.5.2 in the original version), the title of section 3.4.1 (3.5.1 in the original version) needs to be kept.

P18, L17-19: I would suggest to skip over the sentences "Huang et al. (2014) : : :: : : The aerosol in Wuxi close to Shanghai is expected to have similar properties". There are many kinds of properties for aerosols. It is not clear what properties of aerosols are referred to here. The statement that the aerosols in Wuxi have similar properties with those in Shanghai is very speculative.

We agree that there are many kinds of properties for aerosols. Thus we specified the statement as "The aerosols in Wuxi (which is close to Shanghai) are expected to be similarly dominated by secondary aerosol". We prefer to keep it because the previous study motivated our study here. Because the distance between Wuxi and Shanghai is 130km. And the life time of aerosols in the boundary layer is up to several days. The aerosols can be mixed well in the region including Wuxi and Shanghai through a transport of 1-2 days. Thus the dominant aerosol sources can be expected to be

similar.

P19, L18-21: Too speculative.

We think it is reasonable to say that the wind dependence of HCHO indicates the anthropogenic sources of HCHO. Because the natural sources are rather homogenously spread around the measurement site, but the factories which emit VOCs are located in specific areas.

P20, L2: Change "3 km" to "4 km".

Corrected

P21, L18-24: This paragraph can be omitted.

Because we decide to keep section 3.4.2 (see the reply to the second specific comment), the relevant paragraph is needed in the conclusion.

P22-30: Use indented lines for each reference.

Corrected

P31, Table 1: The format of this table looks not good and needs to be rearranged. Try to avoid using the same items in both column and row. For instance, use species for each column, and for the row use Cross section, Fitting interval, Polynomial degree, Intensity, and so on.

We modified the table based on the suggestion.

P34, L9: Change "mean maps of" to "maps of mean".

Corrected

P37, Fig. 5; P39, Fig. 8; P44, Fig. 15: Better to reduce the absolute maximum/minimum values in Y-Axis appropriately so that the differences can be seen more clearly.

We re-plotted the figures as Fig. 6 and 9 in the revised version.

P39-41: Figs. 9-12 can be merged into one figure, with 4 columns for different seasons.

We merged the Figs. 9-12 into one figure (as Fig. 6 in the revised version).

P41-42, Fig. 13: It might be difficult to understand the top panels (colored) of this figure if one had not read the manuscript carefully. It can be more helpful if some words like "primary sky" and "secondary sky" are added, e.g., to the legend, in the figure.

We modified Fig. 13 (Fig. 7 in the revised version) based on the suggestions.

P45-46: It is suggested to move Figs. 16-19 to the Supplement.

We replace Figs. 16-19 by the new Fig. 9 in the revised version (see general comments above).

P48-49, Figs. 21-22: These two figures can be omitted or be moved to the Supplement. Since the seasonal variability of wind directions is not so high, you may consider making a wind rose diagram averaged for all the experiment period and adding it to Fig. 1.

We modified them based on your suggestion.

P58, Fig. 27: The positions of the characters in X-Axis need to be adjusted.

Corrected

P59-60, Fig. 28: This figure can be omitted or be moved to the Supplement.

Because we decide to keep section 3.4.2 (see the reply to the second specific comment), the relevant figure (Fig. 15 in the revised version) is needed to show.

Supp.-P1, L31: With what do NO2 and O4 dSCDS show a systematic increase or decrease?

With the increase of SZA. We correct the sentence.

Supp.-P3, L12: The dSCDs shown here read not as large as two times of the mean RMS.

We corrected the text: They are assumed as two times of the mean RMS.

Supp.-P6, L16: Fig. S9 should be renumbered as its position be moved the place after Fig. S23. Change "the for elevation angles" to "the elevation angles".

We correct it as "the AODs for elevation angles of" in the revised version.

Supp.-P7, Fig. S1; P8, Fig. S2; Fig. 10, Fig. S5: Both RAA and SAA are used. Please check if they refer to the same variable.

We change all "RAA" as "SAA". Because the instrument is pointed to the north, the SAA is the same as RAA.

Supp.-P12, Fig. S8: I did not find a place in the main manuscript as well as in the Supplement that this figure is referred to.

We corrected the manuscript. Fig. S8 (Fig. S17 in the revised version) is cited in section 2.2.3.

Supp.-P15, Fig. S10: This figure should be moved to the front.

We moved it to the front.

Supp.-P19, L3: Change "ds" to "DoF"?

Corrected

Supp.-P7-37: Please try to let the main body figure and its caption to be in the same page.

We followed the suggestion.

Supp.-P39: Use indented lines for each reference.

Corrected

---

## Author Response (AR2)

Dear editor,

We submitted the revised version of our manuscript.

We replied to the comments of the Referee and the Co-editor (see our detailed answers point by point listed below) and made modifications with respect to the comments.

The changes applied to our manuscript are described in detail below. They are also highlighted in the Marked-up manuscript version with 'track changes' (main text and supplement)

With best regards,

Yang Wang

**Reply to Referee**

First of all we want to thank this reviewer for the constructive and helpful suggestions.

General comments
Most of the referees comments have been addressed in the revised manuscript and the presentation quality regarding the number of figures is improved. However, the quality of the English used in the revised or new parts of the manuscript is very poor to my opinion. Although there will be corrections by ACP in the proofreading stage, I think the manuscript should be corrected before its acceptance for final publication.

Author reply:
We paid some efforts to improve the English based on the suggestions from the reviewer and co-editor.

**Specific Comments:**

1) Page 6, line 2 : I would replace 'Tropospheric vertical profiles (in the layer from the ground to the an altitude of 4 km) of aerosol extinctions and trace gases volume mixing ratios are retrieved from the SCDs by a use of the PriAM algorithm' by 'Tropospheric vertical profiles of aerosol extinctions and trace gases volume mixing ratios are retrieved from the ground surface to 4km altitude by using the PriAM algorithm'
Author reply:
We modified the sentence as "Tropospheric vertical profiles of aerosol extinctions and trace gases volume mixing ratios from the ground up to 4km are retrieved from the SCDs by using the PriAM algorithm"

2) Page 7, line 15: 'averaging kernels (AKs)' -> 'AKs'
Author reply:
Corrected

3) Page 8, line 5 : 'and can be very different at different locations' ->'and strongly depends on the location'

Author reply:
Corrected

4) Page 9, lines 20-21: I would replace 'Here it should be noted that AERONET Taihu station is located in a more remote area (from the downtown Wuxi) than the MAX-DOAS at Wuxi station. The different locations could contribute' by 'Here it should be noted that AERONET Taihu station is located in a more remote area (downtown Wuxi) than the MAX-DOAS instrument. This could contribute'

Author reply:
We modified the sentence as "Here it should be noted that AERONET Taihu station is located in a more remote area in Wuxi city than the MAX-DOAS instrument. This could contribute"

5) Page 9, line 23: 'implied' -> 'illustrated'
Author reply:
Corrected

6) Page 10, line 6: 'for different seasons' -> 'for the different seasons'
Author reply:
Corrected

7) Page 10, lines 8-9: 'For the comparisons of AODs between from the MAX-DOAS and the AERONET sun photometer' -> 'For the comparisons of AODs between MAX-DOAS and AERONET sun photometer'
Author reply:
Corrected

8) Page 10, line 32: 'In spring, the worst correlation is found which might be related' -> 'The worst correlation is found in Spring, which might be related'
Author reply:
Corrected

9) Page 11, line 8: 'The higher R' -> 'The higher R value'
Author reply:
Corrected

10) Page 12, lines 1-4: I would replace 'Another point which needs to be clarified is that distinguishing "low aerosols" and "high aerosols" is based on the colour index observed by MAX-DOAS. Thus there is not an explicit AOD value which distinguishes both aerosol categories.' by 'It should be also noted that the distinction between "low aerosols" and "high aerosols" conditions is based on the colour index measured by the MAX-DOAS instrument, and not on explicit AOD threshold values'.
Author reply:
Corrected

11) Page 13, lines 18-21: I would replace 'This is especially true for the retrieved AOD. Especially large mean difference and worse correlation are found under "continuous clouds" for AOD (Fig. 9), and also the cloud effects on the retrieved AE profiles are significant (Fig. 8a)' by 'This is especially

true for the retrieved AOD for which large mean difference and worse correlation are found under "continuous clouds" (see Fig. 9). The impact of clouds on the retrieved AE profiles is also significant, as illustrated in Fig. 8a'.

Author reply:

Corrected

12) Page 13, lines 24-25: 'are found in Fig 8c, e and g' -> 'are found, as can be seen in Figs. 8c, e, and g'

Author reply:

Corrected

13) Page 13, line 26: 'are found' -> 'are obtained'

Author reply:

Corrected

14) Page 13, line 30: 'conditions in Fig. 9' -> 'conditions, as shown in Fig. 9'

Author reply:

Corrected

15) Page 13, line 34: 'under' -> 'for'

Author reply:

Corrected

16) Page 13, line 34: 'imply the degraded performance' -> 'imply a degraded performance'

Author reply:

Corrected

17) Page 14, line 1: 'contribute' -> 'contributes'

Author reply:

Corrected

18) Page 14, lines 3-5: I would replace 'In this study, we decide to keep the MAXDOAS results under "high aerosol" condition in the following analysis because in spite of their lower accuracy they still provide important information.' by 'In this study, we decide to keep the MAXDOAS retrievals under "high aerosol" condition because, despite of their lower accuracy, they can still provide useful information.'

Author reply:

Corrected

19) Page 14, lines 27-30: I would replace 'The estimations of aerosol relevant errors are rough. A further studies need to be done to acquire a more reasonable estimation by considering aerosol

properties, profiles of aerosols and TGs and observation geometries.' by 'These estimations are based on a linear propagation of the aerosols errors on the TG retrievals, which is a rough assumption. Additional sensitivity tests considering uncertainties on aerosol properties and profiles should be performed for the different viewing geometries in order to derive a more realistic error estimate.'

Author reply:

Corrected

20) Page 21, line 21: 'It indicates' -> 'It indicates that'

Author reply:

Corrected

21) Page 24, line 17: I would replace 'Although this study is local and rough, it still shows several general and important results:' by 'Although local, this study shows several general important features:

Author reply:

Corrected

**Reply to co-editor**

Dear Andreas,

Thanks for accepting our manuscript and your many efforts to improve the study. We replied the comments from the anonymous reviewer point-by-point. We modified the manuscript based on the comments and corrections from you and the anonymous reviewer. Please find my replies to your comments in the following:

**Specific comments**

1) please use the same y-scales in Figs. 6 and 9 (with the exception of number of points)
Author reply:
Modified

2) please indicate what has been subtracted from what in the figure captions of figs. 6. 9, and 10
Author reply:
Modified

3) please replace "hour of a day" by "hour of day" in Fig. 14
Author reply:
Modified

4) check formatting of table 1
Author reply:
Modified

5) please check the use of low aerosols / high aerosols in text and figure captions. This is confusing to the reader as what you mean is high / low aerosol load but could be misunderstood as aerosol located low / high in the atmosphere. Also please do not use plural for aerosol here.
Author reply:
We modified the text and figures. Plural for aerosol is not used for the marks of sky condition categories. And we also clarified the point as "Please note that "low" and "high" in the marks of sky condition categories describe the aerosol load but not the aerosol height." in section 2.2.5.

6) please move figure S30 back in the main text as you discuss it so the reader should find it in the manuscript itself
Author reply:
Moved back

7) Personally, I do not see what your discussion of the geometric approximation adds to what is already in the literature. The shortcomings of this approximation are known and the separation of the differences into two terms does not add more information in my opinion
Author reply:
We agree with you on that the shortcoming of the geometric approximation is well known. However our discussion aims to understand the differences of tropospheric VCDs between derived from the geometric approximation and from the profile inversions. Because both sides can contribute to their differences, finding the dominant side can help select a reasonable method to acquire realistic VCDs.

8) In the discussion of Fig. 9 you state that under high aerosol situations, the deviations increase for all trace gases. However, for $NO_2$ this is not the case - on the contrary, for $NO_2$ this is the only scenario where correlation and slope are good!

Author reply:

We agree that this statement is not appropriate. We can't say the deviations of profile retrievals increase under "high aerosol" condition. It is reasonable to say that outlier values of the mean differences and slopes under "high aerosol" conditions are found for aerosol and TG retrievals. We modified the discussion as "In comparison with other sky conditions except "fog" and "thick clouds", outlier values of the mean differences and slopes under "high aerosol" conditions are found for both aerosols and TGs, as shown in Fig. 9. This phenomenon could attribute to two factors: a degraded performance of aerosol profile retrieval and a reality of vertical distributions of aerosols and TGs. On the one hand, aerosol profiles corresponding to high aerosol loads could hardly be well reproduced by the retrieval algorithm due to the constraint of the a-priori profile and the assumption of Eq. (1). This is indicated by the systematic overestimation of the modelled $O_4$ dSCDs compared to the measured $O_4$ dSCDs as shown in Fig. 7 for "high aerosol" conditions. On the other hand, a "high aerosol" condition usually indicates a polluted period, therefore it is reasonable to expect a very different inhomogeneity of the horizontal and vertical distributions of aerosols and TGs in the lowest layer (0-200m) of MAX-DOAS profile retrievals under "high aerosol" conditions compared to other relative clean conditions. Thus different air mass observed by a MAX-DOAS instrument and other techniques could play a different role under "high aerosol" conditions. Further studies on the evaluation and improvement of profile retrievals of MAX-DOAS under heavy aerosol pollution conditions need to be carried out in the future. In this study, we decide to keep the MAXDOAS retrievals under "high aerosol" condition because, despite of their lower reliability, they can still provide useful information. "

9) The editor pointed out the last sentence in section 2.3 is not clear

Author reply:

This sentence aims to answer a comment from reviewers. The reviewer think the discrepancy of retrieved profile from the reality can impact $Diff_{geometry}$ and $Diff_{inversion}$. The approach of separating errors of the geometric approximation and profile retrievals will break down. However we think the reviewer is wrong. $Diff_{geometry}$ is not impacted by the unrealistic retrieved profile. Only $Diff_{inversion}$ is impacted. We modified the sentence to make it clear.

**Marked-up manuscript version:**

[revised manuscript text omitted]